

# Assessment of Short-medium Term Intervention Effects Using CAESAR-Lisflood in Post-earthquake Mountainous Area

Di Wang[1,2,3], Ming Wang[1], Kai Liu[1]

[1]School of National Safety and Emergency Management, Beijing Normal University, Beijing, China.

[2]Academy of Disaster Reduction and Emergency Management, Beijing Normal University, Beijing, China.

[3]Faculty of Geographical Science, Beijing Normal University, Beijing, China.

*Correspondence to*: Ming Wang (wangming@bnu.edu.cn)

**Abstract**

The 2008 Wenchuan earthquake triggered local geomorphic changes rapidly and gradually and produced abundant materials through external processes. The abundant materials increased the risks of geomorphic hazards (flash floods, landslides, and debris flows) induced by extreme precipitation in the area. To reduce sediment transport present in geomorphic hazards, intervention measures such as dams, levees, and vegetation revetments have been constructed in specified sites.

This study concentrated on the assessment of intervention effects incorporated with various facilities on post-earthquake fragile mountains in the short-medium term. Take the Xingping valley as an example, we used the CAESAR-Lisflood landscape evolution model to simulate three different scenarios including unprotected landscapes, present protected landscapes, and enhanced protected landscapes in 2011-2013. We compared the geomorphic changes and defined two indicators to assess the intervention effects.



The results showed that the mitigation facilities were effective, especially engi-
neering measures that cooperated with vegetation revetments in the upstream area,
and the present mitigation measures were inadequate to stop materials loss and pre-
vent hazards from the upstream area. Moreover, the effectiveness reduced gradually
caused by the storage capacity of dams decreased. The simulation methods assessed
the ability and effectiveness of cooperated control measures and could support opti-
mum mitigation strategies.
## 1. Introduction
Strong earthquake shaking fractures rock mass; the resulting cracks are propa-
gated into a weak plane (Huang, 2009) by weathering and erosion; the resulting
source materials increase in mountainous regions, and modify mountain landscapes by
various surface processes for days, years, and millennia (Fan et al., 2019). That means
the quake-stricken areas will trigger landslides (a general term to describe the
downslope movement of soil, rock, and organic materials under the influence of grav-
ity and also the landform that results from such movement) by complicated processes.
The devastating earthquake measuring Ms =8.0 (the surface-wave magnitude which is
the logarithm of the maximum amplitude of ground motion for surface waves with a
wave period of 20 seconds) that struck the Wenchuan area has produced landslides
that threaten highways, railways, towns, and other infrastructure. Although limited
comprehensive mitigation measures were constructed in potentially dangerous sites,
disasters still occurred because of complex processes and origins, high-frequency pre-
cipitation, and the low cost of treatment (Cui et al., 2013; Yu et al., 2010). Therefore,
understanding intervention measures is crucial for effective mitigation. More studies
mainly focus on the establishment of post-evaluation effectiveness index systems


without more practices (N. Wang et al., 2015; L. Zhang and Liang, 2005) and long-
term measurement of changes before and after mitigation measurement by field sur-
veys (Chen et al., 2013; Zhou et al., 2012). The subjective expression determines that
the index system establishment is still in the theoretical stage and the measurement
cost is high in time and money. Recent research compares the disaster characteristics
before and after the intervention, which are quickly obtained from disaster simulation
(Cong et al., 2019). While the characteristics express the process ignoring the long
time effects on the geomorphic changes. Thus, the short-medium term and spatial geo-
morphic changes quickly obtained from the simulation will provide more details to in-
terpret engineering measures in special sites even in those inaccessible to humans.

The open access 2-D landscape evolution model CAESAR-Lisflood (C-L) is

based on Cell Automata (CA) framework, which has powerful spatial modeling and
computing capabilities to simulate complex dynamic systems (Batty et al., 1997;
Batty and Xie, 1997; Couclelis, 1997), enables the study of many earth system inter-
actions with different environmental forces. Representation of deposition and erosion
within C-L is used widely in rehabilitation planning and soil erosion predictions from
a post-mining landform (Hancock et al., 2017; J.B.C.Lowry et al., 2019; Saynor et al.,
2019; Slingerland et al., 2019; Thomson and Chandler, 2019) and channel evolution
and sedimentary budget in dam settings (Gioia and Schiattarella, 2020; Poeppl et al.,
2019). In addition, there have been a series of studies in the mountainous area involv-
ing secondary geo-hazard driving factors (Li et al., 2018; M. Wang, Liu, et al., 2014)
and vegetation recovery (X. Zhang et al., 2018). C-L was used with different scenarios
of rainfall or future climate change to interpret the landscape evolution after the Wen-
chuan earthquake (Li et al., 2020; Xie et al., 2018). The methods and parameters val-
ues used in the above research helped to promote the application in other study areas.





In this study, we compared the short-medium term scenario simulations to assess
the effectiveness of a set of mitigation facilities and to analyze the role of each meas-
ure in the specific site. The results will guide the control of secondary geological dis-
asters after an earthquake.
**2.Study area**
2.1 *Regional characteristics*
The study area was Xingping valley in the northeastern Sichuan province, a left
branch of the Shikan River (a tributary of the Fu River) (Fig. 1). There are nearly two
hundred households scattered among more than five villages in the catchment. The to-
pography of the catchment is rugged with an elevation between 800 and 3036 m and
an area of approximately 14 km$^2$. The catchment shape looks like a "leaf" with a
nearly U-shaped main ditch characterized by a high longitudinal gradient (~ 120‰)
and more than ten small V-shaped branch gullies. The length from northeast to south-
west is 5770 m, the other direction perpendicular to which is 4150 m. The region is
characterized by a humid temperate climate with a mean annual temperature of
14.7℃. The mean annual precipitation is up to 807.6 mm with maxima between May
and September. The steep terrain and short-term heavy rainfall make an ephemeral
stream in this area.
The geological settings are mainly distributed metamorphic sandstones, sandy
slate, crystalline limestone, and phyllite of Triassic Xikang Group ($T_{3xk}$) and Silurian
Maoxian Group ($S_{mx}$), which easily induce a large amount of loose solid material after
weathering of a static process. Wenchuan earthquake, a dynamic process made this
area one of the most severely affected locations with a Modified Mercalli Intensity
scale of IX and X (M. Wang, Yang, et al., 2014). The earthquake strengthened the


solid material produced and reached $10^6$ m$^3$ by triggering landslides and other surfi-
cial movements from Mayuanzi, Zhengjiashan, and Wujiaping (Fig. 1)(Guo et al.,

2018).

2. 2 *Historical hazards and intervention measures*

To reflect most of the landslides processes in spatial relationships according to

the site survey and literature research on the characteristics of the historical hazard,
we divided the study area into three regions: source area, translation area, and deposit
area (Feng et al., 2017; Guo et al., 2018; Zhao et al., 2019) (the dashed lines in Fig. 1.
(c)). The loose solid materials induced by severe rain are easily motivated from the
source area to the deposit area through the translation area. There burst 6 group debris
flow-flash flood disaster chains in rainfall season according to field surveys. Table 1
shows the occurred time, total rainfall of each period, corresponding disaster descrip-
tion, and landslides distribution delineated from remote sensing image data.

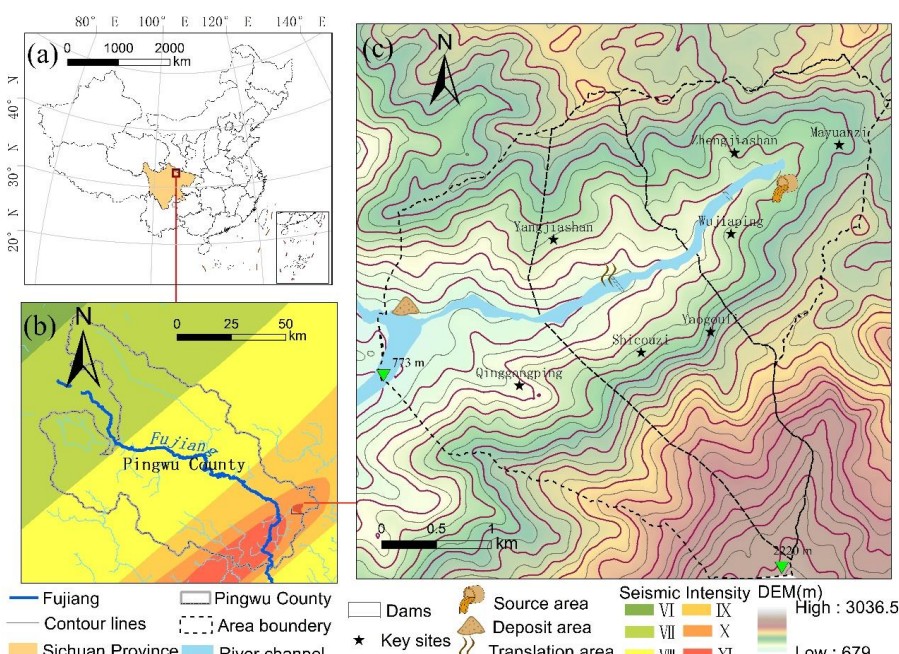

Fig. 1 The location of the study area. (a) Location within China. (b) Location within the seismic intensity ranges of the Wenchuan earthquake. (c) The spatial relationship of the source area, translation area, deposit area, and distribution of elevation.

Table 1 History of hazards in the study area

| Time | Total rainfall (mm) | Details | Landslides distribution |
|---|---|---|---|
| 2008.9.24 | 140.0 | The debris flows after the earthquake first broke out from Mayuanzi and the deposited sediment was up to $5.0\times10^4$ m$^3$ at the junction with the Shikan river, which resulted in collapsed houses and a mess of farmland in the inundation. * | / |
| 2009.7.15-7.16 | 200.0 | The debris flow erupted for 20 min and carried $2.5\times10^4$ m$^3$ solid materials into the outlet section in the catchment. * | / |
| 2010.8.13 | 223.3 | Loose materials were carried from branch outlets into the main outlet and deposited in their routes. * | |
| 2011.8.20 | 118.0 | The scenario was like in 2010.8.13, while damaged less. * | |




| | | | |
|---|---|---|---|
| 2013.7.7-7.12 | 800.0 | The landslides occurred in the upper steep branch, turning to a rapid and large flow-like motion in the main outlet and sweeping over the houses, pigsty, and arable land near the channel. Eventually, the mixture of soil and fragmented rocks accumulated $29.5 \times 10^4$ m³. * | |
| 2018.7.9-7.11 | 360.0 | Several branches burst debris flows, and the materials from Qing-gangping accumulated on the road more than 2 m. * | |

*means the sources are mainly from literature research (Feng et al., 2017; Guo et al.,
2018; Zhao et al., 2019)

120   Vulnerability to landslide hazards is a function of a site's location (topography,

geology, drainage), type of activity, and frequency of past landslides (Highland and
Bobrowsky, 2008). Consequently, this landscape will not stop experiencing landslide
hazards in the short term. To stabilize the loose solid materials, an engineering control
project was completed in October 2010. The project included two blocking dams, one
of which was in the upper source area and the other in the translation area (Feng et al.,
2017)(Fig. 1(c)). The storage capacity of the two reservoirs are, respectively, $5.78 \times 10^4$
m³ and $7.2 \times 10^4$ m³ and the upper dam (10.0 m) is higher than the other one (9.0 m).
With deposited in the reservoirs gradually, the first dredging work was after landslide
hazards in 2013 and the upper reservoir remained at half capacity in 2016, meanwhile,
the lower reservoir was full of loose material.

## 3. Materials and Methods

132  *3.1 Scenarios settings*

133   The abundant source materials of landslides are provided in the quack-stricken

area. Control processing should be performed to prevent the transportation of loose
solid materials. We simulated three scenarios incorporating engineering measures and
biological measures to assess the geomorphic response in 2011-2013 and then as-


sessed the effectiveness of intervention measures. Scenario UP: Unprotected land-
scapes, which means the sediment will move with no anthropogenic intervention. Sce-
nario PP: Present protected landscapes, the present two blocking dams stop a large
amount of material from moving downslope in 2011-2013 without dredging work all
the time (see section 2.2). Scenario EP: Enhanced protected landscapes, the two
blocking dams in Scenario PP plus vegetation revetments in the source area and lev-
ees in the deposit area. The placement of additional facilities was decided by the an-
nual field survey results, where there are still a large number of materials and the set-
tlements would be damaged every year (see Fig. 2 and Section 3.2.2). The vegetation
revetments reduce erosion by enhancing the infiltration capacity of soil and reducing
the surface flow velocity. The levees are artificial embankments to protect the plow
land and buildings; they are constructed to prevent flow and prevent the mix of water
and sediment from overflowing and flooding surrounding areas. We simulated and
compared the three types of situations described above.
*3.2 C-L model description and setting*

The C-L (Tom J Coulthard et al., 2013) was integrated the Lisflood-FP 2D hy-

drodynamic flow model (Bates et al., 2010) with the CAESAR geomorphic model (T
J Coulthard et al., 2002; Van De Wiel et al., 2007), which is based on CA framework
to suit the gridded data required in geomorphic processes simulation. Its stronger
physical basis in a two-dimensional hydrodynamic flow model and faster simulation
in a complete catchment over time scales from hours to thousands of years made it our
surface process simulator. The catchment mode requires the surface digital elevation
model (DEM), the bedrock DEM, the grain size distribution, the rainfall data and
other parameters (Table 2), and related output settings.





161 Besides the creative flow model, which is used to simulate the shorter term hy-

162 drodynamic effects, there are three main parts hydrological model, erosion and depo-

163 sition model, and slope progress. The hydrological model uses input rainfall data to

164 generate runoff in the catchment based on adaption of TOPMODEL (Topography

165 based hydrological model) (Beven and Kirkby, 1979), which is routed in flow model

166 including velocity and depth, which are then used to calculate shear stress that can

167 then be used to calculate fluvial erosion and deposition. The slope model enables ma-

168 terials from the slope to be fed into the fluvial system with mass movement occurring

169 when a critical slope threshold is exceeded and soil creep as a function of the slope.

170 These models update variables in square gridded cells at any time interval, such as el-

171 evation and derived topographic data, grain sizes and proportion data, hydrological

172 data (e.g., discharge, water depth, velocity), and other types of generalization data.

173 For three different scenarios, we reconstructed four parameters formatted differ-

174 ently in catchment mode such as DEM, bedrock DEM, M, and rainfall series. The ar-

175 rangements of the input parameters are described as follows.

176 3.2.1 Surface and bedrock digital elevation model

177 Although the run time of the C-L simulation increases exponentially as the num-

178 ber of grid cells increases, to describe clearly the control process, especially the two

179 dams and levees in the catchment, we unified grid cell scales to 10 m for all needed

180 data. The GlobalDEM product with a 10 m × 10 m resolution and 5 m (absolute) ver-

181 tical accuracy was used as the prepared data to form three types of initial DEMs (UP

182 DEM, PP DEM, and EP DEM). Before rebuilding initial DEMs, we filled the sinks of

183 the original GlobalDEM, which were prone to form by interpolation operation, and

184 then caused the hydrological module to calculate inconsequently. The DEM could be

185 used as the surface DEM of the unprotected landscapes (UP DEM) in 2011. According



to the engineering control project described in Section 3.2.2, present protected land-
scapes' surface DEM (PP DEM) was added 10 m and 9 m in the location of dams, re-
spectively. Similarly, the enhanced protected landscapes' surface DEM (EP DEM) was
extracted by increasing the value of specified grid cells which would be expressed
levees building based on PP DEM. And the height of the levees was 2 m, an average
height used in the lower river channel of the study area to prevent high and fast flow.
From the field survey and the contents of section 2.2, the spatial distribution of
erodible thickness (the difference between surface DEM and bedrock DEM) was dif-
ferent. The bedrock DEM included in this model for each scenario to stop eroding was
derived by subtracting erodible thickness from surface DEM.  The distribution of
erodible thickness was divided into five regions (Fig. 2) by comparing the foundation
of buildings, the exposed bedrock, and the residents' memory of the history of land-
slides deposited. The average thicknesses of upstream low and high-altitude areas
were set to 10 m and 3 m, respectively, and the erodible layer in the downstream area
was supposed to be 3 m. For the river channel and outlet, there would be a large
amount of deposition and there were supposed to be 5 m and 4 m approximately. The
engineering control processes with two dams in Scenario PP and two dams cooperated
with levees in Scenario EP were supposed to be non-erosive concrete. So the erodible
thickness of the engineering control processes area was 0 m. Fig. 2 shows the flow
chart of the generation of DEMs and bedDEMs. In addition, all of the DEM were for-
matted ASCII raster as required in C-L.


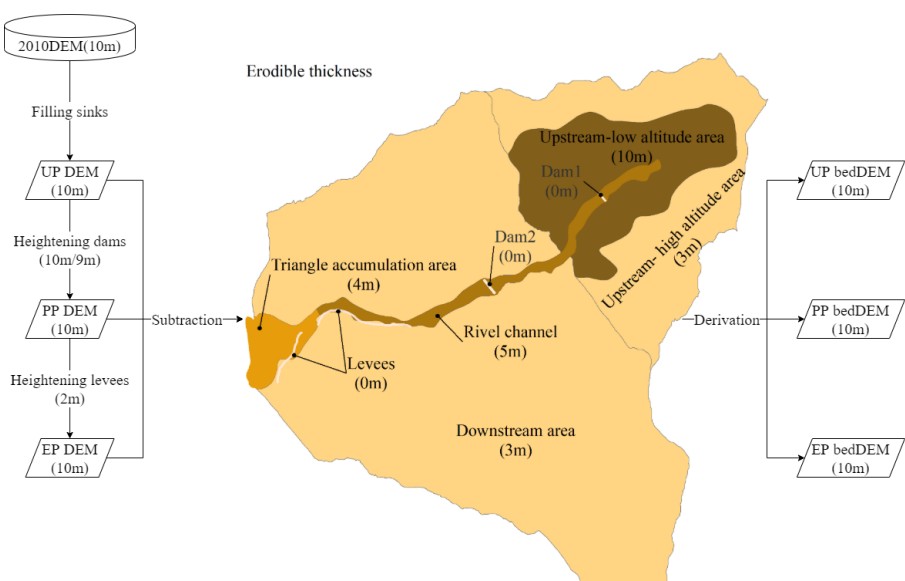


Fig. 2 Flow chart describing the generation of DEMs and bedDEMs (bedDEM: bedrock DEM). All the
numbers attached to DEM on both sides indicated the DEM grid's width and the numbers under facilities such as
dams on the left one are height measured from surface DEM. The numbers in central erodible thickness are the
depth of the material which is capable to remove by runoff.
3.2.2 Vegetation settings
Another parameter in scenarios used in simulations was "m" which controlled
the exponential decline of transmissivity with depth (Batty et al., 1997) and influ-
enced the peak and duration of the hydrograph in response to rainfall. The lower the
value of "m", the lower the vegetation coverage, the flashier flood peaks, and the
shorter duration hydrographs. In this research, the "m" in UP and PP scenarios were
set to 0.008 without spatial variation, which represented that the vegetation coverage
is similar to farmland referenced to research in the same study area by Li et al.,
(2020). As mentioned earlier, the upstream-low attitude area covered by the biological
measures designed in the EP scenario indicated a high value of "m". To distinguish
the "m" in the biological protected area clearly, the "m" was set to 0.02, equal to the
vegetation coverage in the forest (Li et al., 2020).
3.2.3 Rainfall data
In this research, we compared three scenarios with identical precipitation data
during 2011 and 2013 as mentioned in section 3.1. The source data of precipitation in
2011-2013 (Fig. 3(a)) was from the China Meteorological Administration
(http://data.cma.cn) with daily temporal resolution. The rainfall intensity and the fre-
quency of extreme events affect patterns of erosion and deposition (Tom J. Coulthard
et al., 2012), therefore, we used the stochastic downscaling method to generate hourly
data to best capture the hydrological events in this study, which was introduced by Li
et al., (2020) and Lee and Jeong, (2014). The referenced hourly precipitation was
from the pluviometer located 20 km from the study area in 2016(Fig. 3(b)), with an-
nual total precipitation of 684 mm. The rainfall in 2016 was characterized by (1)
hourly precipitation from 1.1 mm to 35.4 mm and (2) the maximum and average dura-
tion of a rainfall event up to 24 h and 2.8 h. In the downscaling method, the daily rain-
fall was divided into four levels (>100 mm, 50-100 mm, 20-50 mm, and 0-20 mm)
and the referenced hourly rainfall series of those days whose daily rainfalls were close
to the value on the day at a certain level were combined by reproduced, crossed and
mutated included in the genetic algorithm (Goldberg, 1989). At last, the downscaled
rainfall series were generated by gathering the normalized hourly data based on the
daily rainfall. Fig. 3(c) shows the downscaled rainfall series in 2011-2013, which il-
lustrated that the downscaled hourly precipitation series was better than the hourly-
mean precipitation (5.27 mm) in the day with maximum precipitation (126.5 mm).

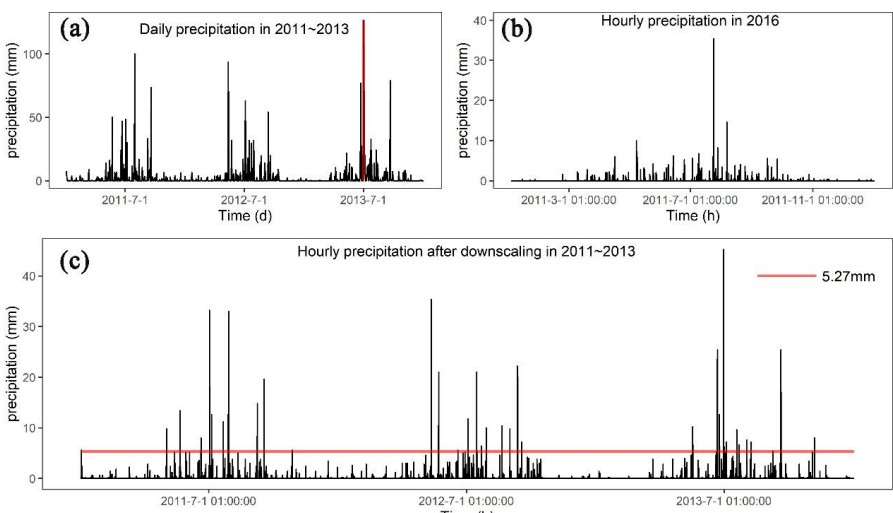


Fig. 3(a) showed the required downscaling daily precipitation in 2011-2013 (the red vertical line indicates daily
maximum precipitation of 126.5 mm); (b) showed the referenced hourly precipitation in 2016; (c) showed the
downscaled hourly precipitation in 2011-2013 (the red horizontal line indicates the hourly-mean precipitation 5.27
mm in the day with maximum precipitation showed in (a))
3.2.4 Other parameters
The C-L model is sensitive to a set of model physically based parameters in-
cluded in Skinner et al., (2018) for an identical catchment with a grid cell size of 10 m,
such as slope for edge cells, grain size set, vegetation critical shear stress, and Man-
ning's n values. These parameters were determined by the application of Xie et al.,
(2018) and Li et al., (2020) in the same study area. In particular, the manning n rough-
ness was set according to suggested values (Arcement and Schneider, 1989) in differ-
ent land-use, and other more sensitive parameters were determined by repeated experi-
ments such as the minimum Q value (see Table 2).
Table 2 The C-L parameter values for the simulations of three different scenarios.

| Parameters | Value | Description |
| --- | --- | --- |
| 9 kinds of grainsizes (m) (grainsize proportion) ★★ | 0.000074(0.098), 0.0005(0.138), 0.001(0.052), 0.002(0.162), 0.005(0.158), 0.01(0.169), | Used for calculating the sediment transport in each active layer |


| | | |
|---|---|---|
| | 0.02(0.13), 0.04(0.06), 0.1(0.033) | |
| Suspended fall velocity(m/s) | 0.0003 | Designated as the falling velocity for the finest fraction(74μm) |
| Sediment transport formula ★★★★ | Wilcock and Crowe | A criterion calculated the fluvial erosion and deposition for all cells |
| Max erode limit (m) ★★★ | 0.002 | The maximum amount of material that can be eroded within a cell at each time step |
| In channel lateral erosion rate ★★★ | 20 | Controlling the channel narrowing |
| Active layer thickness (m) | 0.1 | The thickness of a single active layer |
| Lateral erosion rate ★ | 0.000003 | The variable controls lateral erosion |
| Lateral edge smoothing passes | 40 | The number of passes for the edge smoothing filter (distance between two meanders) |
| Vegetation critical shear stress (Pa) ★★★ | 100 | The value above which vegetation would be removed by fluvial erosion |
| Grass maturity rate (yr) ★ | 1 | The speed at which vegetation reaches full maturity in years |
| The proportion of erosion that can occur when vegetation is fully grown | 0.1 | Determined the effects of vegetation maturity on "in channel lateral erosion rate" and the "lateral erosion rate". |
| Soil creep rate(m/yr) ★★ | 0.0025 | The variable tends to cause erosion gradually on sharper features in the terrain |
| Slope failure threshold (°) ★★★ | 60 | Angle threshold in degrees above which landslide occur |
| Input/output difference allowed(m³/s) ★★ | 0.5 | Described the flow model running in a steady state and used to speed up the model operation |
| Min Q for depth calculate(m) ★★★ | 0.1 | The value above which the flow depth would be calculated to save running time |
| Water depth threshold above which erosion will happen(m) | 0.01 | The value above which the model starts to calculate erosion |
| The slope for edge cells ★★ | 0.005 | The exit cells' slope to control the erosion and deposition |
| Evaporation rate (m/d) ★★★ | 0.00418 | Used to calculate the evapotranspiration |
| Courant number | 0.3 | The value controls the numerical stability and speed of operation of the flow model |
| Mannings n roughness (forest, farmland, landslide, river channels) ★★ | 0.02, 0.008, 0.003, 0.002 | The roughness coefficient used by the flow model |

Note: The greater the number of ★, the more sensitive to the model, and the unlabeled parameters were not
studied (Skinner et al., 2018).


*3.3 Output analyses*

The overall temporal and spatial changes in internal geomorphology under three

different scenarios were available to assess intervention measure effectiveness. The
simulated elevation changes on the last day of each year were selected to show the de-
tails, which were derived from the difference between output DEMs and initial DEMs
(EleDiffs). The EleDiffs indicated the depth of sediment deposition or erosion (>0:
deposition, <0: erosion). We classified the depth to show the distribution of the depo-
sition and erosion, defined the total damaged area in each scenario by summing all af-
fected cells' areas, and compared the damaged area of every classification in three
scenarios. In addition, we zoomed in on the key spots including blocking dams, lev-
ees, and vegetation revetments to explore the geomorphic response to various control
measures in different scenarios and record the depth of deposition in dams blocking
areas. To quantify the changes in the internal source area, translation area, and deposi-
tion area, the sediment volumes of deposition and erosion were calculated respec-
tively from the EleDiffs cuboid.

In different scenarios with different intervention measures, the divided regions

would behave differently in sediment conservation. To quantify the conservation abil-
ity conveniently, we defined some related variables based on the sediment balance
system (Fig. 4). In the balance system, for the region $n$, the deposited sediment ($D_n$)
and the input sediment from the upper connected region ($I_n$) are equal to the eroded
material volume ($E_n$) plus the output volume to the next region ($O_n$) in the same pe-
riod. Based on the relationship between variables shown in Eq.1 and Eq.2, we defined
$C_a$ (Eq. 3) to quantify the sediment conservation ability.
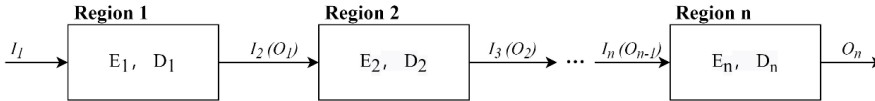





Fig. 4. The sediment balance system in the study area (the Region $n$ indicated source area, translation area, and de-
posit area in this study)

$$I_n = \sum_{2}^{n} E_{n-1} - \sum_{2}^{n} D_{n-1} \tag{1}$$

$$I_n + E_n = O_n + D_n \tag{2}$$

$$Ca = \frac{D_n}{I_n + E_n} \tag{3}$$

Where $n$ is the region number of source area (=1), translation area (=2), and de-
posit area (=3).
The daily sediment yield measured in the valley was the other important output
variable of sediment transport. We referenced a terminology from the stock market in
economics to assess the relative efficiency (Eq. 4, compared with Scenario UP) of en-
gineering protections in scenario PP and engineering cooperation with biological
measures in scenario EP.

$$Re_{PP/EP,i} = \frac{Q_{UP,i} - Q_{PP/EP,i}}{Q_{UP,i}} \tag{4}$$

Where $i$ is the sequence of day; $Q_{UP,i}$ is daily sediment yield volume from the
outlet in Scenario UP of day $i$; $Q_{PP/EP,i}$ is daily sediment yield volume from the outlet
in Scenario PP or Scenario EP of day $i$; $Re_{PP/EP,i}$ is daily relative effectiveness of con-
trolling measures in Scenario PP or  Scenario EP of day $i$.
4. **Results**
4. 1 *Overall geomorphic changes*
There were three panoramas at the end of each year in each scenario, which were
classified into seven ranks by natural breaks for EleDiffs (Fig. 5): extreme erosion (-
15-10 m), heavy erosion (-10--7 m), moderate erosion (-7--3 m), light erosion (-3--1


m), micro change (-1-1 m), light deposition (1-3 m), moderate deposition (3-7 m),
heavy deposition (7-10 m), and extreme deposition (10-14 m). The erosion and depo-
sition aggravated in a similar spatial pattern in all three scenarios. Erosion occurred
mainly in the upper reaches of the main channel and the branches on both sides,
among which the left branches were extremely serious, such as Qinggangping gully
and Shicaozi gully. As shown in Fig. 5, the three scenarios appeared to have different
distribution patterns, especially around the two dams. Statistically, the Scenario UP
damaged 0.76 km$^2$ (5.4% of the total catchment), the PP affected 0.70 km$^2$ (5.0% of
the total catchment), and the EP decreased the area to 0.61 km$^2$ (4.4% of the total
catchment). The damaged area reduced gradually as the more controlling measures for
loose solid materials.

In addition, the affected area of seven ranks changes showed different effects in

the different scenarios. Erosion in the three scenarios was similar in that the light and
the moderate erosion areas were more than the area of extreme and heavy erosion.
The area of each erosion degree in UP was almost larger than that in PP and both
larger than that in EP, except that the light erosion area in PP was slightly larger than
that in the UP. For the deposition in the internal geography, the greater deposition
depth, the less coverage of deposition. Especially, the extreme deposition area was
slightly more than the area of the heavy deposition in UP. Further analysis showed
that the extreme, moderate, and light deposition areas decreased to varying degrees in
the order of UP, PP, and EP, and the heavy deposition areas showed the opposite trend,
which mainly contributed to the blocking of dams and vegetation revetment.

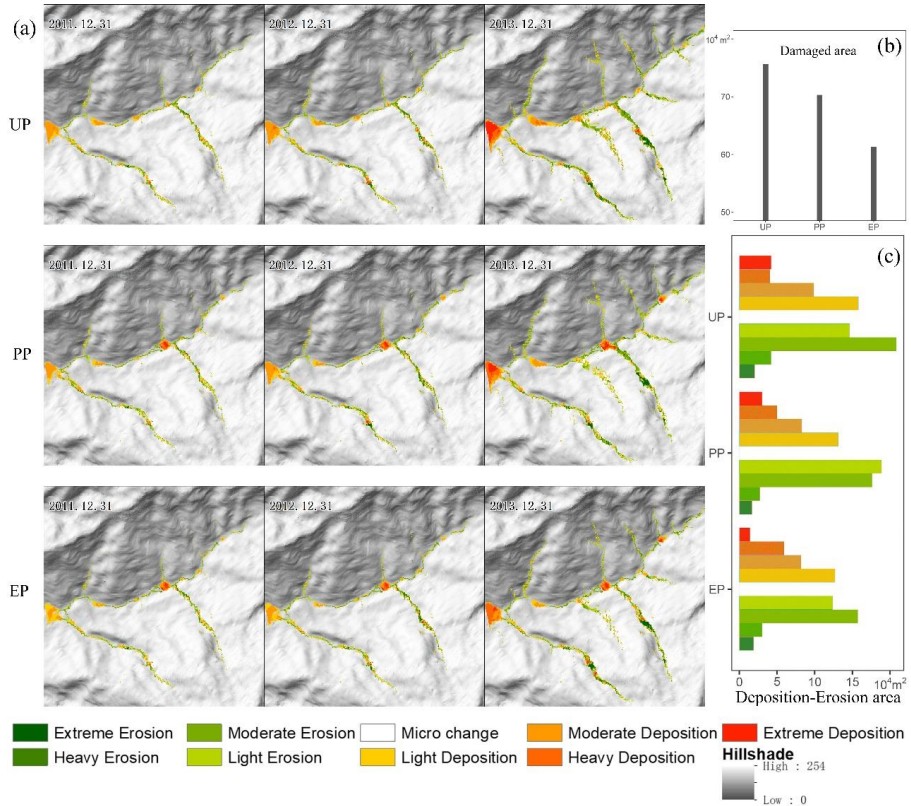

Fig. 5. (a) Simulated internal geomorphic changes over time for three scenarios; (b) the damaged area included deposition and erosion for three scenarios; (c) the final different ranks of deposition and erosion for three scenarios.

### 4. 2 *Details of key spots*

An amplified investigation of the controlling measures around their position detailed the differences in the three scenarios. Therefore, the upriver land panorama, containing two dams in Scenario PP and cooperating with extra biological measures in Scenario EP, was used to outline the affected area, measure the impacts on blocking sediment, and examine how the biological area helped to stabilize the slope. Similarly, the panorama of downriver land described the two levees in scenario EP escape from the debris, protecting the plow lands and buildings.



In the upriver reservoirs of the two dams (Fig. 6), the evident orange clusters in-

dicated the accumulation in Scenario PP and EP, whereas erosion showed in green in

the scenario UP. The area of accumulation blocked by dam 1 in EP was similar to PP's

area while the accumulation in EP covered a larger range than that in PP blocked by

dam 2. Further analysis (Fig. 7) about the depth of deposition blocked by two dams

showed that the depth blocked by dam 2 was larger than that blocked by dam 1 in

Scenario UP and PP. Whereas, the deposition depth blocked by dam 2 decreased to be

slightly lower than that by dam 1 in Scenario EP. In Scenario PP, the sediment depth

blocked by dam 1 was larger than the height of the dam body at the end of simulation

time. Similarly, the accumulation blocked by dam 2 exceeded the dam height at last.

In Scenario EP, both the reservoir areas of dam 1 and dam 2 were lower than the

dams' height.

The materials produced from upriver tributary gullies varied in three scenarios by

the extra biological protection measures. There yielded $14.4 \times 10^4$ m$^3$ loose materials

in EP's biological protection area (solid lines in Fig. 6). In the same gullies, the loose

materials were $27.1 \times 10^4$ m$^3$ and $16.9 \times 10^4$ m$^3$, respectively in Scenario UP and PP

without biological protection. The vegetation revetment enhanced the sediment con-

servation based on the role of dam 1. In addition, the materials were carried mainly

from the two gullies in the upriver of dam 2 and the downriver of biological protec-

tion area, which was inferred from the larger amount of erosion volumes in two gul-

lies in each scenario ($48.2 \times 10^4$ m$^3$, $42.5 \times 10^4$ m$^3$, and $35.2 \times 10^4$ m$^3$ in Scenario UP,

PP, and EP).


In the downriver area, the levees had an important role in preventing debris and
protecting the property. Compared with the accumulation in UP and PP without lev-
ees, the levees in EP blocked debris in the bend of the channel and protected the resi-
dents and cultivated land along the river.

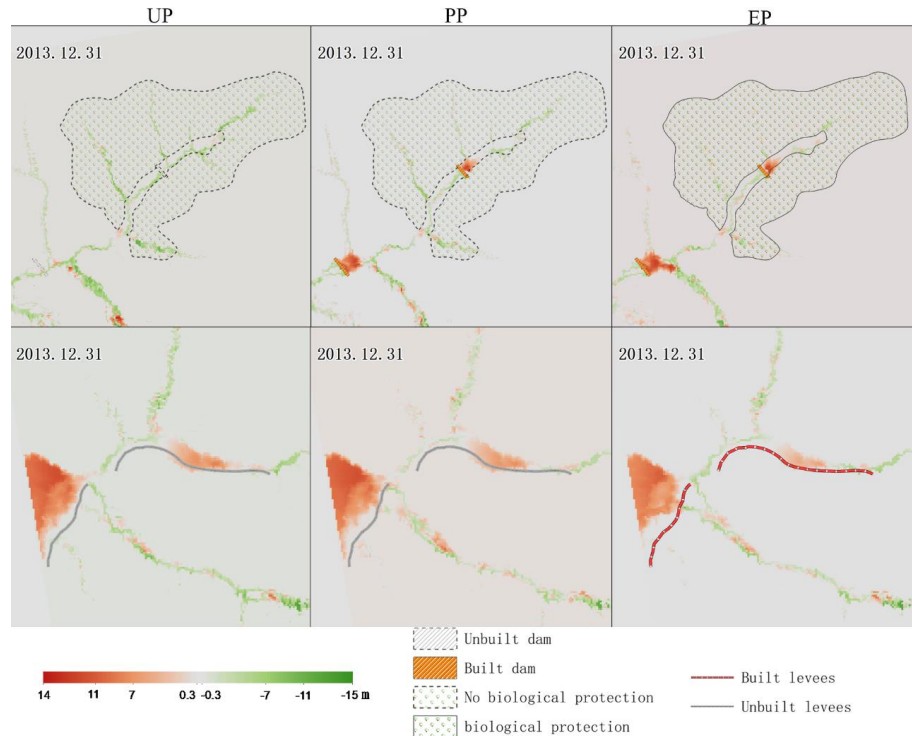


Fig. 6. The final detailed geomorphic changes in the key spots (the upriver dam 1, dam 2 in Scenario PP and
EP and vegetation revetment in Scenario EP showed in the first row; the downriver levees in Scenario EP represent
in the second row)

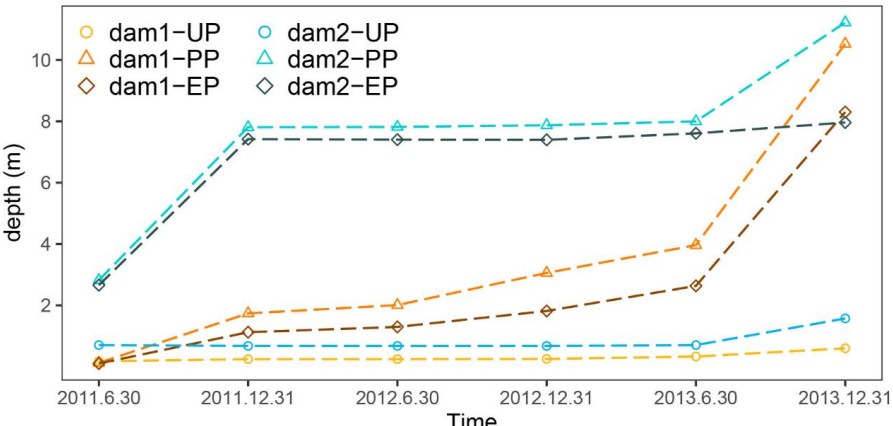


Fig. 7. The depth of deposited sediment blocked by dams in three scenarios
4.3 *Divisional erosion and deposition*

We analyzed the source area, translation area, and deposit area by calculating the

eroded and accumulated sediment volume. Fig. 8 shows the erosion and deposition
distribution induced by rain over three years. The data showed similar phenomena in
three scenarios, i.e., the eroded volume in the source area was less than that in the de-
posit area, and both were less than that in the translation area. The degree of deposi-
tion in the source area was less than that in the translation area, and the largest deposi-
tion was in the deposit area.

From the analysis of sediment conservation ability (see section 3.3) in each re-

gion controlled by different measures in three scenarios, the deposit area was the best
at all times, and the source area was the worst. Dam 1 in the source area and Dam 2 in
the translation area were so effective that the materials conservation ability increased
by 138.1% and 52.5% in Scenario PP compared with Scenario UP, respectively (Table
2). What's more, the mitigation measures with vegetation revetment and levees in
Scenario EP worked better. The ability in the source area increased by 161.9%, and
the levees helped increase by 3.49% compared with Scenario UP. Therefore, the dams


were most effective in blocking sediment, the vegetation revetment strengthened the
conservation ability, and the levees worked mainly to prevent damage.

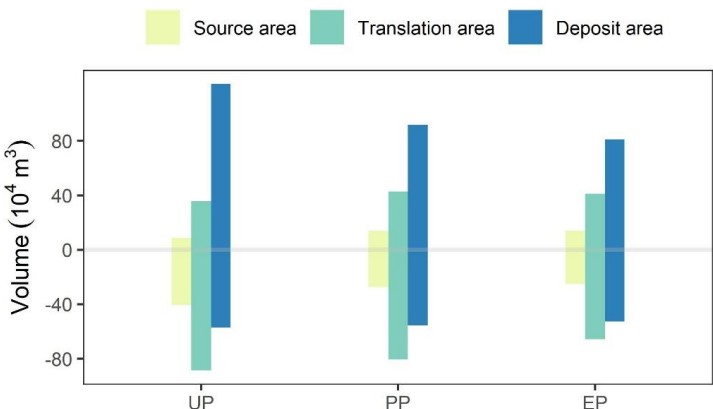


Fig. 8. The deposition and erosion volumes in different areas

Table 2 The sediment conservation ability

| Region            Scenario | UP | PP | EP |
|---|---|---|---|
| Source area | 0.21 | 0.50 | 0.55 |
| Translation area | 0.40 | 0.61 | 0.62 |
| Deposit area | 0.86 | 0.86 | 0.89 |


4.4 *Effectiveness assessment*

Fig. 9 presents the time series of cumulative sediment yield for each scenario ac-

cording to the output file. The steep curve means the great increase of sediment and
three increasing stages have high consistency with the rainfall intensity in three mon-
soons (May-September). The total sediment output in UP was the largest, about
$30.4 \times 10^4 \ m^3$, and the total output in PP ($26.3 \times 10^4 \ m^3$) was larger than that in EP
($19.3 \times 10^4 \ m^3$). We used the formula mentioned in section 3.3 to calculate the relative
efficiency of controlling measures by human intervention in PP and EP (Fig. 9b).
Three distinct stages were clear for the effective degree between PP and EP. The stage





I showed that the two dams in PP or two dams with two levees and vegetation protec-
tion in EP both controlled the sediment loss. Later stage II was an existing and pecu-
liar period where the effect of enhanced protective measures in EP was not as good as
that in PP after many simulation trials. In stage III, the relative efficiency of the inter-
vention measures in EP was greater than that in UP, which could achieve long-term
effective and stable conservation of solid materials. What's more, the relative effi-
ciency values in PP's stage III showed a decreasing trend, whereas the values declined
indeterminately in EP's stage III because of the slight increase in values at the end of
the simulation. In general, the engineering works in controlling sediment transport
were efficient, and it would be better at protecting the fragile environment effectively
with other intervention measures like vegetation revetment and levees. In addition, the
effectiveness of conservation and mitigation would decrease with time.

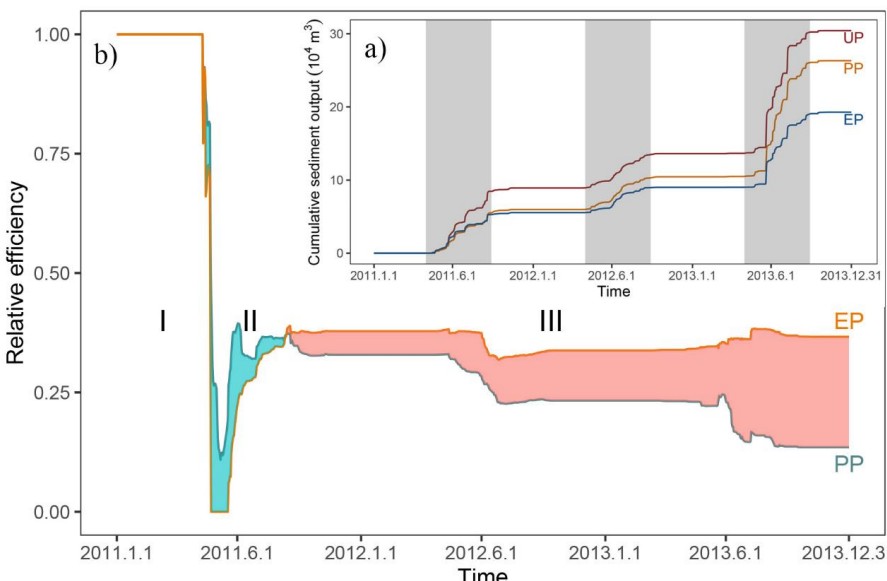


Fig. 9. a) showing the output cumulative sediment over time (grey region highlighting three monsoons); b)
showing the relative efficiency of scenario UP and EP compared with the UP (green region representing PP more
effective than EP and red region standing the opposite)


## 5. **Discussion**

### 5.1 *Reliability and uncertainty*

Reliability and uncertainty deserve a discussion for understanding and imple-
menting the simulation results in most geographical analyses and modeling processes
(Yeh and Li, 2006). Comparative simulation tests using the C-L tool suggested a com-
plex spatial and temporal evolution of sediment transport. In addition, the tool demon-
strated that the efficiency according to space and time varied in scenarios, which dif-
fered in control measures conducted on the mountainous areas that are susceptible to
secondary geo-hazards. In this study, for the parameters involving geological condi-
tions, we cited local research and comprehensive parameter sensitivity papers; we
downscaled the daily rainfall sequence into hourly rainfall data collected in 2016 for
every year because the total rainfall and intensity were identified as 'normal year' rain-
fall in 2016 (Xie et al., 2018). For the generated input data, although the intensity and
event time would not be the same as the actual value, the realization of total rainfall in
three different years suggested reasonable differences.
In addition, the optimal simulation result was decided according to the sediment
depth in dam reservoirs and output between simulation and actual measurement from
field survey or literature research. Fig. 10a shows the sediment distribution blocked
by dam 1 in August 2012; the distance from the dam crest to the deposition level was
up to 7 m, which suggested that the buried dam depth was nearly 3 m (dam height: 10
m). Therefore, the 3 m-depth simulation result of PP in the same moment found in
Fig. 6 (see section 4.2) was consistent with the actual value. In October 2013, the
same location collected by photo in Fig. 10b showed that the reservoir was full of ma-





terials, which were equal to the simulation depth of more than 10 m in Fig. 6. Con-
versely, the sediment yield in 2013 was up to $29.5 \times 10^4 \text{m}^3$ (Feng et al., 2017), which
was from mainly the Shicouzi gully. Coincidentally but more scientifically, the appar-
ent new erosion that occurred in 2013 in Shicouzi (Fig. 4) suggested the disaster his-
tory was rebuilt successfully by simulation, and the erosion volume in Shicouzi was
$20.6 \times 10^4 \text{ m}^3$. Therefore, it was reasonable that the simulation of eroded materials
from Shicouzi accounted for 70% of the sediment from the left branch gully.

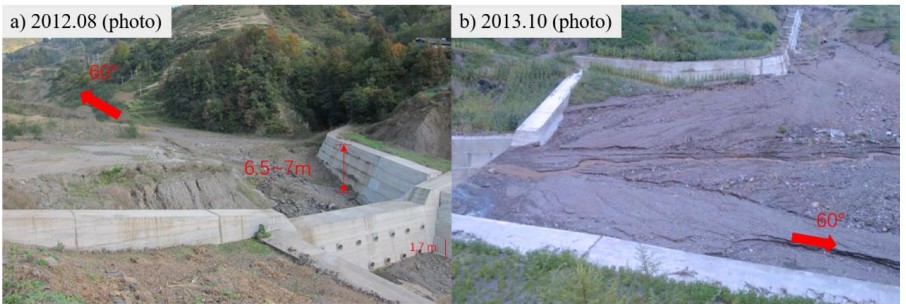


Fig. 10. The photos of dam 1 reservoir (the red single arrows showing the azimuth angle and the double ar-
rows showing the height of the dam body)
5. 2 *Short-medium term problem*
We used an ingenious and simple method to build the dams and levees in the
simulation by increasing the elevation in the expected location and assuming that it
could not be eroded (see https://sourceforge.net/projects/caesar-lisflood/). This
method proved to be experimentally feasible (Gioia and Schiattarella, 2020; Poeppl et
al., 2019). The rigid dam and levee body embedded in the model would not be broken,
and the effect would not be weakening, so the result of geo-hazard risk assessment
would be reduced to some extent. Although the fast and large amount of moving de-
bris triggered a tremendous impact in the simulation, the tools could not simulate the


461 geo-hazard chain links and would ignore the fierce attack on the environment and fa-

462 cilities downstream. Some typical geo-hazard chains were focused on the specified

463 event in a short time and recreated the hazard lifecycle using physical and mechanical

464 models (Fan et al., 2019). We concentrated on the effectiveness of mitigation

465 measures in the short-medium term, which is different from those in space-time scales

466 and purposes. Therefore, the three-year simulation time made it underestimated risk

467 assessment, and a success to simulate the effect of mitigation measures compared with

468 the actual result in this study.

469 5.3 *Sediment transport patterns*

470  Different from the typical debris flow research, where three divided areas get

471 their names for the materials process, the simulation result demonstrated the loose

472 solid materials from the source area sliding to the resting area were the least among

473 the three regions, even for the scenario UP (unprotected landscapes). The sediment

474 transport patterns change considerably and two reasonable descriptions are as follows.

475 First, the abundant loose solid materials formed by the strong earthquake have stabi-

476 lized generally since 2008's debris flow (details in Table 1). Second, the long, deep,

477 and steep gullies are mainly located in the translation area (Yaogouli, Shicouzi,

478 Yangjiashan) and deposit area (Qinggangping). Thus, the large erodible area and the

479 poor topographic conditions destroyed the circulation and deposit area more than the

480 source area. Just as Fig. 11 shows, the movement of the materials occurred mainly in

481 the branches in the circulation and deposit area.
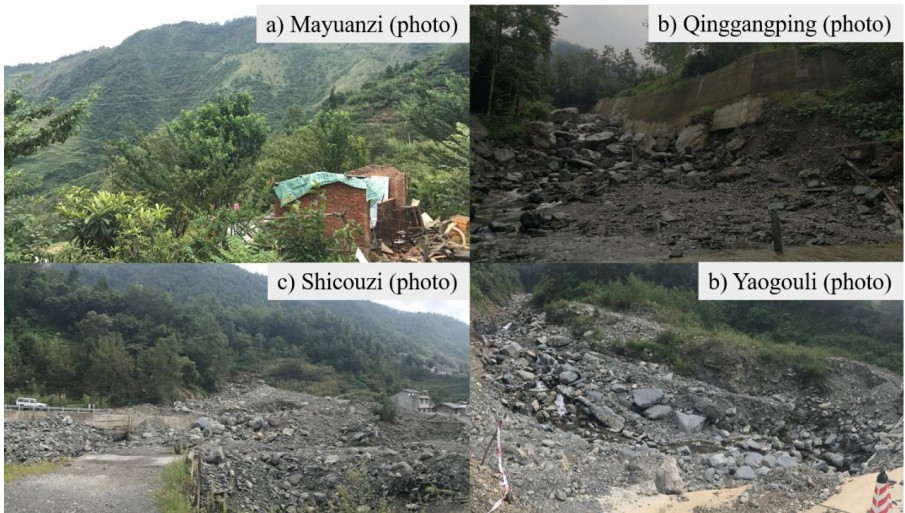

Fig. 11. Photos showing hazards sites in different areas: a) the source area, b) the deposit area, c) and d) the translation area

## 5. 4 *Long-term trials*

In the future warmer world with more water vapor in the atmosphere, precipitation extremes will be intensified, increasing the likelihood of extreme and intense rainfall (East and Sankey, 2020). Then sequential increased fluvial transport capacity and erosion would accelerate geomorphic changes. With increased uncertainty of precipitation and temperature, future work about landscape evolution of three scenarios will help to understand long timescale effectiveness of intervention measures. We randomly selected one of the 50 repeat datasets downscaled by Li et al., (2020), which were generated in 2013-2025 and RCP 4.5 emission scenario from NEX-GDDP (spatial resolution: 0.25°×0.25°, temporal resolution: daily) to simulate the effectiveness in three scenarios. The result (Fig. 12) illustrated that stage III (stable stage started on the 161st day, in which Scenario EP's intervention measures were more effective) was more than stage I and II, which were only in the beginning. The relative effectiveness




in both scenarios decreased gradually and the curve went down faster in PP than that
in EP.

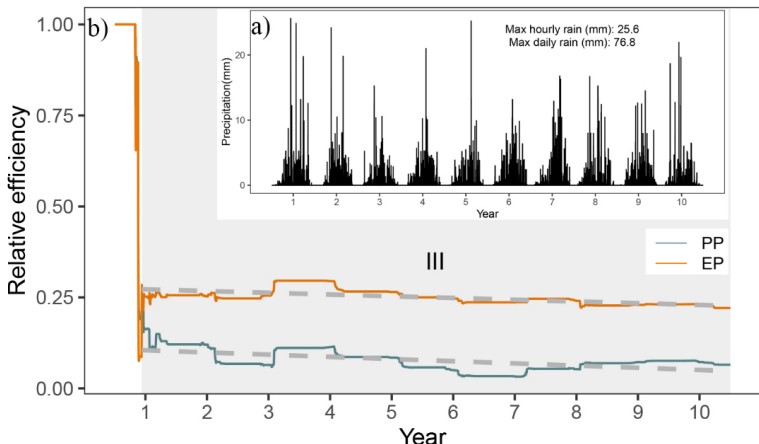

Fig. 12. a) Rainfall downscaled from stochastic future rainfall; b) the relative efficiency changes over ten
years (grey region highlighting stage III, and the grey dashed lines indicated the linear fitting curve)
6. **Conclusions**
We have four key findings. First, the comparative scenario simulations showed
that mitigation measures in scenario PP (containing two blocking dams) and scenario
EP (incorporating biological processing in the source area with two dams) were effec-
tive in reducing erosion, controlling sediment output, and protecting property from
damage in post-earthquake fragile mountains prone to secondary geo-hazards. Erosion
had a high consistency with the monsoons (May-September) and was mainly in the
upper reaches and the left branches of the main gully. The two dams have blocked the
upstream sediment successfully and the levees had an important role in preventing the
debris shocking, and burial of the residents and cultivated land along the river. In ad-
dition, the decrement in EP suggested the accumulated materials blocked by dams up-
grade a slope upstream in turn. What's more, model embedded quantification of vege-



tation revetment showed that the sediment yielded decreased 5 times as much as sce-
nario UP, which contributed to that the vegetation cover enhanced precipitation infil-
tration and reduced flow velocity.

Second, reasonable and comprehensive treatment methods for a mountainous

area with abundant solid materials reduced internal geomorphology changes and sedi-
ment output. The areas of erosion and deposition varied in degree decreased in EP
compared with PP, except for heavy deposition. Then both the internal damaged area
and the erosion volume in EP were less than in PP. In addition, the reduced volume of
erosion in the source area between EP and PP was larger than the deposition volume
suggesting the vegetation protection was effective in EP. Conversely, three years later,
the simulated depth of accumulation blocked by dam 1 and dam 2 was greater than the
height of the dams in PP, whereas only the depth deposited in the upriver of dam 2
was greater than the dam height. Moreover, the present intervention measures are not
adequate to reduce erosion and should be combined with dredging work.

Third, zonal statistics of the volumes of erosion and deposition in the source

area, translation area, and deposit area demonstrated that the characteristics of sedi-
ment transport patterns changed considerably. The conservation ability in the deposit
area was the best at all times, and the source area was the worst. Dam 1 in the source
area and dam 2 in the translation area worked so well that the materials conservation
ability increased by 138.1% and 52.5% compared with the scenario without any inter-
vention method. With the extra help of vegetation revetment, the ability in the source
area increased by 161.9%, and the levees helped the deposit area increase by 3.49%.

Fourth, the two types of effectiveness found in the sediment output simulated un-

der Scenario PP and EP compared with Scenario UP were divided into three apparent
stages with a general downward trend. The first stage was completely effective in both



PP and EP, whereas stage Ⅱ was a peculiar period in which the effect in EP was not
as good as that in PP, which would be caused by the increasing complexity of the
model. Lastly, steady effectiveness would be sustainable as shown in stage Ⅲ, in
which the effectiveness simulated in EP with vegetation revetment and levees was
greater than that in PP.
Taking long-term effectiveness and the function of vegetation into consideration
for mitigation measures is more helpful to understand the efficiency. More works
should be carried out to explore, especially with the increased likelihood of extreme
and intense rainfall in the future.
**Declaration of interest statement**
The authors declare that they have no known competing financial interests or per-
sonal relationships that could have appeared to influence the work reported in this pa-
per.
**Author contribution**
Di Wang: Conceptualization, Methodology, Software, Writing-original draft prep-
aration. Ming Wang and Kai Liu: Supervision, Methodology, Writing- Reviewing and
Editing, Validation.
**Acknowledgments**
This research was supported by the National Key Research and Development
Plan (2017YFC1502902). The financial support is highly appreciated. The authors
would also like to thank Professor Tom Coulthard and his team for their excellent
work on the freely available C-L model (https://sourceforge.net/projects/ caesar-
lisflood).



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
