# Peer review of "An Assessment of Short-medium Term Interventions Using CAESAR-Lisflood in a Post-earthquake Mountainous Area"

_Natural Hazards and Earth System Sciences, 2022_

## Referee Comment (RC1)

In this study a landscape evolution model, CAESAR-Lisflood (CL), is applied to a steep mountain catchment to assess the effectiveness of engineering works in reducing the transport of sediment. This is an applied study, that is straightforward, and demonstrates the use of CL in a highly dynamic landscape. Overall, the manuscript fits the scope of NHESS and would be interesting to modellers and practitioners working in mitigating geo hazards in mountainous regions. My concerns with the study are related to the choice of hydrological parameters, the physical plausibility of landscape changes, and development of initial conditions. In addition, the clarity of the manuscript requires substantial improvement and I recommend the text is thoroughly edited by a native English speaker, with a background in fluvial geomorphology, before acceptance. Below are major comments that need to be addressed followed by a list of minor points and edits.

Major comments

A weakness of the study is the lack of calibration of the hydrological component in CL. As such, there is no way of knowing if the quantity and timing of the floods in the ungauged catchment are accurately replicated by CL. The hydrological parameters adopted (m-values) are from studies performed from nearby catchments but these studies also have not performed calibration to derive m-values. The authors, instead rely on landcover to assign m-values, but m-value is only partly dependent on landcover. For example, Ramirez et al.2022 found that in a mountain catchment soil depth correlated well with m-value and not with landcover. To have greater confidence in the model, the authors need to provide hydrographs for the entire simulated period and, in addition, provide qualitative or quantitative data that confirms the physical plausibility of the simulated discharge, specifically the floods.

In this study, CL simulations have produced locations of deep erosion between 3-10 m in a period of three years. This is quite a bit of erosion in such a short period and in some instances would produce features in the simulated landscape that resemble small canyons. Could you verify that these erosional features are physically plausible by providing photographic evidence from the observed landscape and compare them to cross-sections from the simulation. Or provide any other type of validation that supports such extreme erosion across the simulated landscape. In addition, across all simulations (Fig. 5a), there are instances of erosion that exceed 3 m in the downstream area where erodible thickness is 3 m. Can the authors explain how simulated erosion can exceed the thickness of the initial erodible sediment? Likewise, in this study, how is it possible for CL to produce erosion between 10 and 15 m, if the maximum depth of erodible sediment in the catchment is 10 m?

In the study there is no mention of establishing initial conditions by spinning-up the model to mix the grain sizes. If spin-up was not performed, can the authors provide an explanation. If spin-up was performed, could you briefly explain how it was done in the methods. Regarding the choice of bedrock elevation (Fig. 2), could the authors provide the physical basis for the choice of erodible thickness values and locations of these values.

Modelling the long-term geomorphic response to check dam failures in an alpine channel with CAESAR-Lisflood. International Journal of Sediment Research, 37(5), 687–700. https://doi.org/10.1016/j.ijsrc.2022.04.005

Minor points and edits

Line 20: replace Take with Taking

Line 29-30: "Moreover, the effectiveness reduced gradually caused by the storage capacity of dams decreased." needs rewording.

Line 46: be more specific than just stating "complex processes and origins"

Line 47: "treatment" doesn't sound right

Line 50: "without more practices" doesn't sound right

Line 52: "The subjective expression" has not been defined, so most readers won't know what this means

Line 56-57: define "long time effects" and "short-medium term" is it 10s, 100s, 1000s years?

Line 59: what is meant by "special sites"?

Line 60: add reference for CL.

Line 61: Cellular Automata

Line 67: add Ramirez et al. 2020 and Peleg et al 2021 as recent examples of simulating channel evolution with CL

Ramirez, J. A., Zischg, A. P., Schürmann, S., Zimmermann, M., Weingartner, R., Coulthard, T., & Keiler, M. (2020). Modeling the geomorphic response to early river engineering works using CAESAR-Lisflood. Anthropocene, 32, 100266. https://doi.org/10.1016/j.ancene.2020.100266

Peleg, N., Skinner, C., Ramirez, J. A., & Molnar, P. (2021). Rainfall spatial-heterogeneity accelerates landscape evolution processes. Geomorphology, 390, 107863. https://doi.org/10.1016/j.geomorph.2021.107863

Line 68: add Ramirez et al. 2022 as another recent example for applying CL to dams

Modelling the long-term geomorphic response to check dam failures in an alpine channel with CAESAR-Lisflood. International Journal of Sediment Research, 37(5), 687–700. https://doi.org/10.1016/j.ijsrc.2022.04.005

Line 75-77: "assess the effectiveness of a set of mitigation facilities" in doing what? In reducing channel change, sediment transport…be more specific.

Line 98-99: "The earthquake strengthened the solid material produced and reached $10^6\,m^3$" needs rewording

Figure 1: Remove contours to improve clarity of map (besides contours are not labelled and have limited use). "Translation area" should be replaced with "transitional area", and the same should be done throughout the entire manuscript. Dams are barely visible in map, perhaps fill them with a color. Spell out "Figure" instead of using the abbreviation and do this for all captions.

Line 107: replace "motivated" with "transported"

Line 108-109: "here burst 6 group debris flow-flash flood disaster chains in rainfall season according to field survey" needs to be reworded and 6 needs to be spelled out.

Line 110: replace "occurred time" with "time of occurrence"

Line 111: replace "remote sensing image data" with "remotely sensed image". In addition, provide more information about the image, like the name of the satellite, spatial resolution and date of acquisition. Also, provide more information on how you determined the location of the landslides.

Table 1: "mess of farmland" needs to be reworded. Maps in table are quite small, perhaps remove them from the table and make them larger in a multi panel supplemental information figure, and

reference this in the text. In the table you could add the summary statistics of the landslides derived from the maps (e.g. total area, min area, max area).

Line 127-128: replace with "The upper dam has storage capacity of x m$^3$ and a height of m, and the transitional area dam has a storage capacity of x m$^3$ and a height of m"

Line 128: replace "With deposited in the reservoirs gradually," with "With the reservoirs gradually filling with deposits,"

Line 130: what do you mean by "lower reservoir was full of loose material.", be more specific.

Line 133: quake-stricken

Line 134: "Control processing" needs rewording

Line 145-146: Provide an example of a vegetation revetment.

Line 147: replace "plow" with "agricultural"

Line 152: Change to NHESS citation format: (Tom J Coulthard et al., 2013)

Line 152: Change to NHESS citation format: (T J Coulthard et al.)

Line 155-158: This text needs rewording

Line 161: "Besides the creative flow model" needs rewording

Line 163: "slope progress" needs rewording

Line 166: CL has sediment transport equations, provide the use of these function in the model description.

Line 170-172: This text needs rewording

Line 173: "we reconstructed four parameters" is not entirely correct wording because the DEM and bedrock layer are model initial conditions, the rainfall is a model driver, and the m-value is a parameter.

Line 183-184: "which were prone to form by interpolation operation, and then caused the hydrological module to calculate inconsequently." needs rewording

Line 185: Explicitly mention if the UP DEM does or does not contain dams.

Line 186-188: should be reworded to: "the present-day protected landscape surface DEM (PP DEM) included the dams by raising the grid cell elevations by 10 m for the upper dam and 9 m for the dam in the transitional area"

Line 189: replace "extracted" with "produced and "

Line 188-191: should be replaced with "The enhanced protected landscape surface DEM (EP DEM) includes the dams in PP DEM and, in addition, levees were represented at selected locations. Levees were produced by raising grid cell elevations by 2 m."

Line 192: You mention a field survey and how this led you to conclude the thickness of erodible sediment was spatially variable. In the text, elaborate more about the field survey and how it informed the production of the bedrock layers in CL.

Line 194: replace "different" with "heterogeneous" or "variable"

Line 194-195: needs rewording

Line 200: remove "supposed to be"

Line 200-204: should be "For the river channel and outlet, there would be a large amount of deposition and the thickness of erodible sediment was set to 5 m and 4 m respectively. The dams in Scenario PP and dams and levees in Scenario EP were non-erosive concrete. As such, the erodible thickness of these features was set to 0 m"

Line 205: define bedDEMs and "In addition, DEMs were formatted to ASCII raster as required by C-L"

Fig 2.: The flowchart is redundant as the text has already described the process quite well. I like the figure, but without the flowchart and then change the caption accordingly.

Line 213-221: should be like this: "Another parameter set in each scenario was the m value in CL's hydrological model (TOPMODEL), which controls the exponential decline of transmissivity with depth (Batty et al., 1997) and influences the peak and duration of the hydrograph in response to rainfall. The lower the m value, the lower the vegetation coverage, the flashier flood peaks, and shorter flood hydrograph duration (Citation needed). In this research, the m value in the UP and PP scenarios were set to 0.008 without spatial variation, which represents the vegetation coverage of farmland as determined by Li et al., (2020) for a catchment nearby with similar landcover. As mentioned earlier, the upstream-low elevation area covered by the biological measures designed in the EP scenario was assigned a higher m value."

Replace all instances of "m" with m value throughout the manuscript.

Figure 3: Capitalize the y-axis label. Caption should be like this "Fig. 3(a) Daily precipitation in 2011-2013 (the red vertical line indicates daily maximum precipitation of 126.5 mm); (b) Hourly precipitation in 2016; (c) Downscaled hourly precipitation in 2011-2013 (the red horizontal line indicates the hourly-mean precipitation 5.27 mm in the day with maximum precipitation showed in (a))."

Line 227: change "(Fig. 3(a))" to (Fig. 3a), and do the same throughout the manuscript for all references to multi panel figures.

Line 229: Change to NHESS citation format: (Tom J Coulthard et al., 2012)

Line 243-244: I am not sure what you mean by "the downscaled hourly precipitation series was better than the hourly mean precipitation", please reword this sentence.

Table 2: Explain the choice of setting the slope failure threshold so high (60 deg), wouldn't this prevent the occurrence of landslides and lead to deep canyonlike incision?

Line 263: remove "internal" and any use afterwards because internal geomorphology does not mean anything

Line 264: should read "assess the effectiveness of the interventions."

Line 265-266: should read "The simulated annual landscapes were analysed to quantify the geomorphic change, and were derived…"

Line 269-270: Here you mention "damage" but what you are really calculating is exposure because you are not calculating a monetary value. Change the text to mention exposure, and I am assuming that a map of settlements or landcover was used to calculate the exposure, if so, provide this map in the supplemental information. If you didn't use a landcover map, explain what you used to derive exposure.

Line 274: remove "internal"

Line 284: explain what a low and high value means for CA

Line 292: Is this daily sediment yield measured at the catchment outlet? If so, change the text accordingly.

Line 303: replace "panoramas" with "landscapes" and do the same throughout the entire manuscript.

Figure 5: Hillshade does not need a legend. Add quantitative values to the erosion and deposition legend. Place one scale bar in a map. In panel (a), remove all dates in maps and place dates once above the top row of maps. In panel (b), it still isn't clear how damage was calculated, but see my comment above. I think caption (c) should read "the distribution of deposition and erosion at the conclusion of the simulation for the three scenarios."

Line 308: "aggravated" is not the right word

Line 310-311: Qinggangping gully and Shicaozi gully need to be labelled once in Fig. 5 because the reader is not familiar with these locations.

Line 312-315: Again, this is exposure not damage, change the text accordingly.

Line 317-327: The narrative you provide is fine, but you need to provide some values that support your comparisons.

Line 322: "internal geography" does not make any sense

Line 333-334: should be "Here, we provide a detailed investigation of the controlling measures and surroundings for the three scenarios."

Figure 6: Unbuilt dam is not legible, fill with a color. Built dam needs to be a different color than red to provide contrast with areas of deposition. No biological and biological protection fill patterns appear identical and make it difficult to see the landscape changes. Just have no fill pattern. The erosion and deposition legend needs a title like "Geomorphic change (m)" and in the caption mention that negative geomorphic change values are erosion, and positive values are deposition. Add an inset map that clearly defines where the key spots are in the catchment. Add a scale bar(s). Label dam 1 and dam 2. Caption should be "Figure 6. Geomorphic changes at the conclusion of the simulation at key spots for the UP, PP, and EP scenarios. Top row is the upriver section containing dam 1, dam 2 and the vegetation revetment. The bottom row is the downriver section containing levees.

Line 341-351: Add values to support the existing qualitative comparison.

Figure 7. Capitalize the y-axis label. Mention in the caption that scenario UP does not have dams, but deposit depths are provided where dams existed in the other scenarios.

In Figure 6, the EP map shows the levees blocking a tributary, is this a mistake in the figure or did you really block this tributary in the simulation. Please explain.

Line 372: "Divisional erosion and deposition" doesn't make sense

Line 373-379: Add values to support the existing qualitative comparison.

Figure 8: I am unsure what is plotted here or what type of plot this is. Is this the distribution of simulated geomorphic change after three years in the source, translational, and deposit areas? If so, are these boxplots? If these are boxplots, provide the median, IQR, whiskers and outliers. Add more explanation to the figure caption and the y-axis label should read "Volume of sediment".

Figure 9: caption should read "(cyan shading represents when PP is more effective than EP and red shading represents the opposite)". Be consistent when labelling panels, as here you use a) and in previous figures you use (a). Do this for all of the figures.

Line 408: replace "What's more" with "Additionally"

Line 410: "indeterminately" doesn't make sense

Line 464: Briefly mention the method applied by Ramirez et al. 2022 for check dam failure.

> Modelling the long-term geomorphic response to check dam failures in an alpine channel with CAESAR-Lisflood. International Journal of Sediment Research, 37(5), 687–700. https://doi.org/10.1016/j.ijsrc.2022.04.005

Line 512: "debris shocking" doesn't mean anything

Line 513-514: "In addition, the decrement in EP suggested the accumulated materials blocked by dams upgrade a slope upstream in turn." needs rewording

Conclusion section: Here you need to further summarize your main findings because the current text reads like a repetition of the results.

---

## Author Comment (AC1)

We thank Jorge Ramirez for his review of our manuscript and making the highly constructive comments and suggestions. We are glad to hear our effort to revise the manuscript. The comments and suggestions were formatted in light and dark blue text. The author's response to the major issues is shown below in **black text**.

**General comments**

In this study a landscape evolution model, CAESAR-Lisflood (CL), is applied to a steep mountain catchment to assess the effectiveness of engineering works in reducing the transport of sediment. This is an applied study, that is straightforward, and demonstrates the use of CL in a highly dynamic landscape. Overall, the manuscript fits the scope of NHESS and would be interesting to modellers and practitioners working in mitigating geo hazards in mountainous regions. My concerns with the study are related to the choice of hydrological parameters, the physical plausibility of landscape changes, and the development of initial conditions. In addition, the clarity of the manuscript requires substantial improvement and I recommend the text is thoroughly edited by a native English speaker, with a background in fluvial geomorphology, before acceptance. Below are major comments that need to be addressed followed by a list of minor points and edits.

**Major comments**

A weakness of the study is the lack of **calibration of the hydrological component in CL**. As such, there is no way of knowing if the quantity and timing of the floods in the ungauged catchment are accurately replicated by CL. The hydrological parameters adopted (m-values) are from studies performed from nearby catchments but these studies also have not performed calibration to derive m-values. The authors, instead rely on landcover to assign m-values, but **m-value is only partly dependent on landcover**. For example, Ramirez et al.2022 found that in a mountain catchment **soil depth correlated well with m-value** and not with landcover. To have greater confidence in the model, the authors need to provide **hydrographs for the entire simulated period** and, in addition, provide qualitative or quantitative data that confirms the physical plausibility of the simulated discharge, specifically the floods.

*1.  The m-values*

The m-values in C-L influence the peak and duration of the hydrograph in response to rainfall (Coulthard et al., 2002), which are usually determined by the landcover (e.g., 0.02 for the forest, 0.005 for the grassland) (Coulthard and Van De Wiel, 2017). In our study, we united the value to 0.008 in our smaller catchment (14 km$^2$) in Scenario UP and PP, which resembles the m-value of farmland covered with lower vegetation in the same catchment studied by Xie et al., (2022), Li et al., (2020) and Xie et al., (2018). In scenario EP, the m-value in the vegetation revetments area was 0.02 to distinguish the vegetation coverage. It has been calibrated in the bigger catchment containing our study area (Xie et al., 2018) by replicating the flood event in 2013.

We have read Ramirez et al., (2022) intensively and learned the new and creative method to calibrate components. They determined m-values in the total catchment after perfect simulation results in a sub-catchment according to the soil depth. And we think

It would exist discrepancies between different regions. In our catchment, the vegetation counts more than the soil depth for the m-value, which is caused by the undetermined soil depth. Therefore, it is not the optimal method for our smaller study area distributed larger amount of landslides deposition and river alluvium stem from metamorphic sandstones and sandy slate.

**2. *Calibration**

Admittedly, it is not enough for our calibration work including referencing the parameters from the published research in the same catchment and using the recommended values by model developers. Considering the ungauged basins before 2015, we replicated the flash flood event in July 2018 by C-L to calibrate the hydrological components.

There are no huge differences in geomorphology, channel location, and landcover before 2013 and after 2018 in our catchment found from the field surveys. Based on Scenario PP (with two check dams), we changed the rainfall series to the two-week hourly precipitation in July 2018, which is recorded by the rain gauge 2.5 km away from the catchment placed in 2015. The simulation results (Figure 1c and Figure 1d) showed the erosion and maximum flood depth deposition distributions in Scenario PP on July 15[th], 2018. As shown in Figure 1c and Figure 1d, we selected three locations randomly to compare the simulation results with remotely sensed images and photos. The results (Figure 2) showed reliable results including sediment deposition and the peak flood depth, which indicate that the flash flood event was replicated successfully by the C-L.

[Figure]

Figure 1. The input rainfall series (a and b) and simulation results of the flash flood event in July 2018 (c and d).

[Figure]

(a) modeling results        (b) images in June 2019        (c) photos in September 2018

(after flooding)        (after flooding)

Figure 2. The comparison of the simulation results to images (GF-2 with 8-m resolution) and photos after the flash flood event in July 2018.

**3.  *Verification**

In section 5.1, we confirmed the plausibility of the simulated results using deposition depth evaluated from photos. Herein, we add the discharge of the entire simulation period for Scenario PP. As shown in Figure 3, we compare two types of discharge recorded in published research (Feng et al., 2017; Guo et al., 2018) with those of simulation results to confirm the physical plausibility. And we captured five flood events where the daily precipitation is more than 50 mm in 2013 and the peak discharge was up to 63.6 m³/s.

[Figure]

Figure 3. (a) The simulation discharge in 2011-2013 in Scenario PP; (b) the verification location; (c) the comparison of the simulated to the recorded discharge.

In this study, CL simulations have produced locations of deep erosion between 3-10 m in a period of three years. This is quite a bit of erosion in such a short period and in some instances would produce features in the simulated landscape that resemble small canyons. Could you **verify that these erosional features are physically plausible** by providing photographic evidence from the observed landscape and comparing them to cross-sections from the simulation? Or provide any other type of validation that supports such extreme erosion across the simulated landscape. In addition, across all simulations (Fig. 5a), there are instances of erosion that exceed 3 m in the downstream area where erodible thickness is 3 m. Can the authors explain how **simulated erosion can exceed the thickness of the initial erodible sediment?** Likewise, in this study, how is it possible for CL to produce erosion between 10 and 15 m, if the maximum depth of erodible sediment in the catchment is 10 m?

**4.  *Revised erosional and deposited features**

Many thanks for your reminding and we have to apologize for our mistakes to show the abandoned simulation result, where the input basedDEMs were generated improperly. Now we update the revised figures as shown in Figure 4 and Figure 5, which both show the spatial distribution of erosion and deposition. We correct the description (extreme erosion (<-7 m), heavy erosion (-7--3 m), moderate erosion (-3--1 m), light erosion (-1-0.1 m), micro change (-0.1-0.1 m), light deposition (0.1-1 m), moderate deposition (1-3 m), heavy deposition (3-7 m), and extreme deposition (>7 m)) in section 4.1 and 4.2. We ensure that all the analysis results and input parameters are consistent and from our optimal simulation after checking all the figures, tables and numbers.

In addition, we thank the reviewer for his suggestions about the figure's details.

[Figure]

Figure 4. (a) Simulated geomorphic changes over time for three scenarios; (b) the exposure area included deposition and erosion for three scenarios; (c) the distribution of deposition and erosion at the conclusion of the simulation for the three scenarios.

[Figure]

Figure 5. Geomorphic changes at the conclusion of the simulation at key spots for the UP, PP, and EP scenarios. Top row is the upriver section containing dam 1, dam 2 and the vegetation revetment. The bottom row is the downriver section containing levees.

**5. *Verification of erosion and deposition**

As shown in Figure 6, we verify the erosional and deposited features by providing photographic evidence from the observed landscape and compare them to cross-sections from the simulated results.

[Figure]

Figure 6. The comparison of cross-sections from the simulation results to the photos in the field measurement locations after 2013 in Scenario PP.

In the study, there is no mention of establishing initial conditions by spinning-up the model to mix the grain sizes. **If spin-up was not performed, can the authors provide an explanation.** If spin-up was performed, could you briefly explain how it was done in the methods. Regarding the choice of bedrock elevation (Fig. 2), could the authors provide the physical basis for the choice of erodible thickness values and locations of these values.

*6. Spin-up processes*

Admittedly, we didn't spin up the model to mix the grain sizes. The purposes of the process are to eliminate the 'walls' and the 'depressions' in the cells and avoid the intense erosion in the hill slope in the early run time. Actually, we preprocess the DEMs by filling sink based on Environmental Systems Research Institute's (ESRI's) ArcMap (ArcGIS, 10.8) to eliminate the problematic pixels. Moreover, for our catchment, the fine grains distributed homogeneously both in the hill slope and the channel five years after the strong earthquake. Therefore, we think the huge difference would not exist. However, we will continue to compare the difference in the future work.

*7. Erodible thickness values*

The bedrock elevation (Fig. 2) was evaluated mainly from the published research from the trusted official institutions in China (Feng et al., 2017; Guo et al., 2018) and verified with our field survey. The institutions described in their research according to the prompt and accurate hazard inventory and the UAV survey. It is difficult for us to provide direct evidence like drilling operations and the fine map because of the steep terrain and a large amount of landslides deposition in these post-earthquake fragile mountains.

**Minor points and edits**

The most minor points and edits would be revised in the manuscript directly and some are reply here.

Line 145-146: Provide an example of a vegetation revetment.

Considering the tree roots play an important role in stabilizing the slope and consolidating the soil, the ecological engineering including vegetation revetment was more and more popular in the mountains. For example, the tree planting patterns was studied by Lan et al., (2020). They listed the artificially planted cypress and pines on the slope.

Line 269-270: Here you mention "damage" but what you are really calculating is exposure because you are not calculating a monetary value. Change the text to mention exposure, and I am assuming that a map of settlements or landcover was used to calculate the exposure, if so, provide this map in the supplemental information. If you didn't use a landcover map, explain what you used to derive exposure.

Actually, we only calculate the total of deposited and erodible area in each scenario at the conclusion of the simulation to compare.

In Figure 6, the EP map shows the levees blocking a tributary, is this a mistake in the figure or did you really block this tributary in the simulation. Please explain.

We have checked carefully and ensured that the deep and narrow outlet of the tributary was not blocked by the levees.

Conclusion section: Here you need to further summarize your main findings because the current text reads like a repetition of the results.

Thanks for your suggestion. Admittedly, our conclusion is verbose and we would summarize the conclusion in the revised manuscript.

**Reference**

Coulthard, T. J. and Van De Wiel, M. J.: Modelling long term basin scale sediment connectivity, driven by spatial land use changes, Geomorphology, 277, 265–281, https://doi.org/10.1016/j.geomorph.2016.05.027, 2017.

Coulthard, T. J., Macklin, M. G., and Kirkby, M. J.: A cellular model of Holocene upland river basin and alluvial fan evolution, Earth Surf. Process. Landforms, 27, 269–288, https://doi.org/10.1002/esp.318, 2002.

Feng, W., He, S., Liu, Z., Yi, X., and Bai, H.: Features of Debris Flows and Their Engineering Control Effects at Xinping Gully of Pingwu County, J. Eng. Geol., 25, 2017.

Guo, Q., Xiao, J., and Guan, X.: The characteristics of debris flow activities and its optimal timing for the control in Shikan River Basin Pingwu Country, Chinese J. Geol. Hazard Control, 29, 2018.

Lan, H., Wang, D., He, S., Fang, Y., Chen, W., Zhao, P., and Qi, Y.: Experimental study on the effects of tree planting on slope stability, Landslides, 17, 1021–1035, https://doi.org/10.1007/s10346-020-01348-z, 2020.

Li, C., Wang, M., Liu, K., and Coulthard, T. J.: Landscape evolution of the Wenchuan earthquake-stricken area in response to future climate change, J. Hydrol., 590, 125244, https://doi.org/10.1016/j.jhydrol.2020.125244, 2020.

Ramirez, J. A., Mertin, M., Peleg, N., Horton, P., Skinner, C., Zimmermann, M., and Keiler, M.: Modelling the long-term geomorphic response to check dam failures in an alpine channel with CAESAR-Lisflood, Int. J. Sediment Res., 37, 687–700, https://doi.org/10.1016/j.ijsrc.2022.04.005, 2022.

Xie, J., Wang, M., Liu, K., and Coulthard, T. J.: Modeling sediment movement and channel response to rainfall variability after a major earthquake, Geomorphology, 320, 18–32, https://doi.org/10.1016/j.geomorph.2018.07.022, 2018.

Xie, J., Coulthard, T. J., and McLelland, S. J.: Modelling the impact of seismic triggered landslide location on basin sediment yield, dynamics and connectivity, Geomorphology, 398, 108029, https://doi.org/10.1016/j.geomorph.2021.108029, 2022.

---

## Author Comment (AC2)

We thank Referee #2 for his review of our manuscript and making the highly constructive comments and suggestions. We are glad to hear our effort to revise the

Manuscript. The author's response is shown below in **black text**.

First and foremost, is **the lack of proficiency and fluency in the use of the English language and grammar**, which is consistently poor throughout this paper. This makes it very difficult to comprehend the contents of the paper – for example, it is not clear what the methods were, or how sets of parameters used in simulations were obtained/derived. Because of this, it was also not clear how the results were obtained and what they actually represented, and consequently, whether the resulting discussion and conclusions could be substantiated or supported. Overall, while the aims and objectives could be understood, it was not easy to determine if they had been met. Unfortunately, the author's unfamiliarity with the English language meant that too many sentences were variously incomplete, made no sense, or utilized inappropriate or misspelt words.

Admittedly, the English language in this research is not proficient and fluent. We will revise it carefully in the manuscript.

Some comments and suggestions for improvement:

Rather than try to describe the background to the CAESAR model and how it works themselves, I believe the authors could more clearly and succinctly acknowledge this by referencing existing publications which describe this ie. Coulthard et al 2012.

As currently written, descriptions of specific parameters and methods used in this study and the scenarios modelled are poorly described, or not described at all – for example:

As C-L users, we intended to introduce the background of the C-L and the relevant parameters' definitions to understand the main theories best.

a table of parameters lists values used in simulations, yet **there is no explanation of how or why the values in the table were selected or used** – for example, those representing vegetation parameters (shear stress, age to maturity, proportion of erosion) - were selected and utilized in model simulations for the scenarios in this study.

We admit that some parameters were not introduced in detail, except for the definitions, such as shear stress, age to maturity, and proportion of erosion.

Some sensitive parameters to the model (Skinner et al., 2018) were referenced from some published research in the same catchment, such as the sediment transport formula, grain size and corresponding proportion and the slope failure threshold (Li et al., 2020; Xie et al., 2018, 2022). In addition, we add the calibration of the components in C-L in the first reply. Other parameters in Table 2 are from the

default values recommended by the developers (such as the max erode limit in the erosion/deposition module and the vegetation critical shear stress) in https://sourceforge.net/p/caesar-lisflood/wiki/Home/.

The authors **do not explain why they selected some of the parameters ie why a specific sediment transport equation** was selected. Depending on the sediment transport equation applied, very different model results may occur.

The sediment transport equation was sensitive to the C-L model (Skinner et al., 2018) and we selected the Wilcock and Crowe referenced from published research studied in the same catchment (Li et al., 2020; Xie et al., 2018, 2022). Actually, the Wilcock-Crowe equations are among the more widely used formulae for predicting fractional bed load transport rates in gravel bed streams. They are developed using bed load transport information obtained in laboratory flume experiments with bed material sediments ranging in particle size from 2.83 to 64 mm (Wilcock et al., 2003). Another alternative equation is Einstein-Brown, which is developed by uniform sediment and lightweight materials ranging in sizes from 0.785 to 28.65 mm based on flume data. From the grain sizes in our study area, the Wilcock-Crowe equations would be the better choice.

It is not clear **what rainfall data was used in the simulations** – whether different sets of data were used for different scenarios, or one set was applied across all scenarios. The text about the downscaling rainfall data is simply confusing and does not address this.

We have revised the downscaling method in the third reply. Now we add Table 1 to address the scenarios settings and the input rainfall. We will provide in the revised manuscript at last.

Table 1 The main settings and input for the three scenarios

| Scenario | Descriptions | Period | DEM (resolution) | Rainfall data |
|---|---|---|---|---|
| UP | no anthropogenic intervention | from January 2011 to January 2013 (3 years) | UP DEM (10m) UP bedDEM (10m) | downscaled hourly precipitation in the period (lumped) |
| PP | the present two blocking dams in the upstream without dredging work | | PP DEM (10m) PP bedDEM (10m) | |
| EP | plus vegetation revetments in the source area and levees in the deposit area based on Scenario PP | | EP DEM (10m) EP bedDEM (10m) | downscaled hourly precipitation in the period (spilt) |

As written, it is not clear how results are **obtained or substantiated** from the methods. The authors **make assumptions about the ability of the model to erode that are not supported or substantiated by any evidence**. Specifically, the authors describe how they have attempted to incorporate levees and dams into simulations by simply increasing the elevation in certain areas and not changing other parameters such as particle size. While this reviewer agrees it may temporarily reduce flow, in the longer term, this may well lead to increased erosion

in other areas around the sides of the dam / levee.

We have added the verification in the first reply including the comparison of the simulation results with photographical evidence and the hydrographs.

We incorporated levees and dams into simulations by changing both the surface DEMs and bedDEMs of dams and levees described in Fig.2.

Admittedly, there are many assumptions in our simulation work and we don't consider the changes of particle size before and after the dams and levees. The study of increase erosion around the sides of the dams and levees was been limited in C-L models owing to the unattackable settings near the engineering facilities. Actually, we pay more attention to the short-medium effectiveness of the interventions, which is present in regional features analyzed from the erosion and deposition in the total catchment and the output in the outlet. If possible, we would study from the smaller-scale by using other simulation models or field surveys in the future work.

The authors have demonstrated **a poor use of figures and tables** to support their results. Specifically,

figures variously lack scales or annotations to indicate where the areas of erosion or deposition are (eg figure 5), or where other features (such as dams) referenced in the text are located (eg figure 1).

Thanks for the suggestions. We added the scales in Fig5. in the first time, but they looked odd because the figure is just to show the distribution of erosion and deposition. At last, we decided to remove the scales and other annotations in the newest one.

Some figure captions do not make sense eg figure 2 does not clearly show any chart or process for generating the bedrock DEMs. Some figures do not appear to contain the information described in the text.

Thanks for the suggestions on figures and tables. We would check them carefully and revise them.

Tables are present in the manuscript which are not referenced in the text; different tables share the same number (eg there are 2 tables labelled as table 2); and some tables do not identify what the units in the table represent.

Admittedly, we made a mistake in table labels, we would check again and revised in the manuscript.

Finally, the author's use of **referencing is poor and inconsistent**.

Thanks for the suggestions on references. We have added some new research listed in each reply. We will check again and revise using the standard styles.

References

Li, C., Wang, M., Liu, K., & Coulthard, T. J. (2020). Landscape evolution of the Wenchuan earthquake-stricken area in response to future climate change. *Journal of Hydrology*, *590*(June), 125244. https://doi.org/10.1016/j.jhydrol.2020.125244

Skinner, C. J., Coulthard, T. J., Schwanghart, W., Van De Wiel, M. J., & Hancock, G. (2018). Global sensitivity analysis of parameter uncertainty in landscape evolution models. *Geoscientific Model Development*, *11*(12), 4873–4888. https://doi.org/10.5194/gmd-11-4873-2018

Wilcock, P. R., Asce, M., & Crowe, J. C. (2003). *Surface-based Transport Model for Mixed-Size Sediment Surface-based Transport Model for Mixed-Size Sediment*. *9429*(February). https://doi.org/10.1061/(ASCE)0733-9429(2003)129

Xie, J., Coulthard, T. J., & McLelland, S. J. (2022). Modelling the impact of seismic triggered landslide location on basin sediment yield, dynamics and connectivity. *Geomorphology*, *398*, 108029. https://doi.org/10.1016/j.geomorph.2021.108029

Xie, J., Wang, M., Liu, K., & Coulthard, T. J. (2018). Modeling sediment movement and channel response to rainfall variability after a major earthquake. *Geomorphology*, *320*, 18–32. https://doi.org/10.1016/j.geomorph.2018.07.022

---

## Author Comment (AC3)

We thank **Christopher Skinner** for his review of our manuscript and making the highly constructive comments and suggestions. We are glad to hear our effort to revise the Manuscript. The author's response is shown below in blue text.

In the second Table 2 (there are two) on page 22, the sediment conservation ability in the source area increases from 0.5 to 0.55 between the PP and EP scenarios, yet I don't think there are any additional interventions in this area (levees and vegetation are in the deposit area). I think this may be an effect of the way the authors have applied the spatially varied "m" in the model. In UP and PP, the "m" has a global value of 0.008 and in EP the vegetated areas are given a separate value of 0.02. It isn't stated by the authors but I believe **their rainfall input is catchment lumped (please could the authors confirm)**. My understanding of CL is that for a lumped input it will average all the "m" values and create a single lumped input from it, in this case making the input for the whole catchment less flashy. Alternatively, the authors could **specify two separate rainfall input areas**, one for the vegetated area and one for the rest of the catchment, in effect making two hydrological response units (HRUs) for the model, each with its own input based on the local "m" value. I don't think this needs to be done for a revised manuscript as I doubt it would change their conclusions materially, but it should at least be acknowledged.

Firstly, we are so sorry about the confusion about the tables' labels and we have corrected them. We agree the increase of the sediment conservation ability between the PP and EP scenarios is an effect of the spatially varied m-values in the model.

*1.   the rainfall input*

Actually, the rainfall input in Scenario UP and Scenario PP is catchment lumped. While in Scenario EP, we divided it into two separate but identical rainfall for the regions with different m-values.

Further on "m", I concur with the comments from Jorge that where possible the value should be calibrated against gauged data. If this is not available, basing the value on land cover, as the authors have done, is reasonable. However, the authors are using downscaled hourly rainfall, not observed hourly rainfall, so **any calibration would need to account for this**.

*2.   Calibration*

Admittedly, it is essential to calibrate the hydrological components before replicated work. We follow both of the two reviewers' suggestions and calibrate the parameters by replicating the flash flood event in July 2018 using C-L.
There are no huge differences in geomorphology, channel location, and landcover before 2013 and after 2018 in our catchment found from the field surveys. Based on Scenario PP (with two check dams), we changed the rainfall series to the two-week hourly precipitation in July 2018, which is recorded by the rain gauge 2.5 km away from the catchment placed in 2015. The simulation results (**错误!未找到引用源。**c and

错误!未找到引用源。d) showed the erosion and maximum flood depth deposition distributions in Scenario PP on July 15th, 2018. As shown in 错误!未找到引用源。c and 错误!未找到引用源。d, we selected three locations randomly to compare the simulation results with remotely sensed images and photos. The results (Figure 2) showed reliable results including sediment deposition and the peak flood depth, which indicate that the flash flood event was replicated successfully by the C-L.

[Figure]

Figure 1. The input rainfall series (a and b) and simulation results of the flash flood event in July 2018 (c and d).

[Figure]

|  | (a) modeling results | (b) images in June 2019 (after flooding) | (c) photos in September 2018 (after flooding) |

Figure 2. The comparison of the simulation results to images (GF-2 with 8-m resolution) and photos after the flash flood event in July 2018.

I also concur with Jorge's comment on **spin-up period**. There are no details in the manuscript and it would be helpful to know. In this case, where much of the eroded material is fresh and loose, it could be argued a spin-up might actually be a counterproductive in this instance.

**3. Spin-up period**

Admittedly, didn't spin up the model to mix the grain sizes. The purposes of the process are to eliminate the 'walls' and the 'depressions' in the cells and avoid the intense erosion in the hill slope in the early run time. Actually, we preprocess the DEMs by filling the sink based on Environmental Systems Research Institute's (ESRI's) ArcMap (ArcGIS, 10.8) to eliminate the problematic pixels. Moreover, for our catchment, the fine grains were distributed homogeneously both in the hill slope and the channel five years after the strong earthquake. Therefore, we think a huge difference would not exist. However, we will continue to compare the difference in future work.

The downscaling of the rainfall to hourly is really important (as shown nicely in Figure 3). Sorry to push one of my papers, but Coulthard and Skinner (2016: https://doi.org/10.5194/esurf-4-757-2016) provides some analysis of why and it would be useful to refer to this here. Unfortunately, I found the description of how this was done not clear – please could the authors **revisit this description** so it is easier to follow. It would be useful to also know the spatial resolution of the rainfall product used and the spatial resolution it applied to the model with (I assumed it was lumped).

Thanks for the recommendation. The findings from a range of simulations have revealed that using time-averaged climate inputs may be under-predicting basin

sediment yields as well as introducing spatial biases through under-predicting or over-predicting erosion (Coulthard and Skinner, 2016), which helps to explain why we downscale the daily rainfall into hourly rainfall. Therefore, we would reference it in our revised manuscript.

*4.   Downscaling method description*

The description of the temporal downscaling process was revised as:

In this research, we compared three scenarios using identical precipitation data during 2011 and 2013 as mentioned in section 3.1. The source daily precipitation of one station in 2011-2013 was downloaded from China Meteorological Administration (http://data.cma.cn). The rainfall intensity and the frequency of extreme events affect patterns of erosion and deposition (Coulthard and Skinner, 2016; Coulthard et al., 2012). And we used the stochastic downscaling method to generate hourly data to best capture the hydrological events in this study, which was introduced by (Li et al., 2020; Lee and Jeong, 2014). The referenced hourly precipitation was measured from the pluviometer located 20 km from the study area in 2016, with annual total precipitation of 684 mm. The rainfall in 2016 was characterized by (1) hourly precipitation from 1.1 mm to 35.4 mm and (2) the maximum and average duration of a rainfall event up to 24 h and 2.8 h. The main processes of the downscaling method are as follows.

- extracting the measured daily rainfall closest to the referenced daily rainfall in 2011-2013 through the threshold setting and producing the genetic operators from the extracted hourly rainfall;
- mixing on the genetic operators by genetic algorithm (Goldberg, 1989) composed of reproduction, crossover and mutation and repeating until the distance between the predicted daily rainfall and the measured rainfall is less than the setting threshold;
- normalizing the hourly precipitation to remain the daily rainfall value unchanged.

The input of generated hourly precipitation is catchment lumped in Scenario UP and EP and divided into two separate but identical rainfall in Scenario EP.

A **verification of the model outputs** for the PP scenario by comparing them to real-world observations would strengthen the analysis of the paper. For example, Figure 10 and related discussion could be included within the results as a form of verification for the model outputs.

*5.   Verification*

Similar to the reply on the first review, we add the verification both in the discharge and erosion/deposition features. As shown in Figure 3, we compare two types of discharge recorded in published research (Feng et al., 2017; Guo et al., 2018) with those of simulation results to confirm the physical plausibility. As shown in Figure 4, we verify the erosional and deposited features by providing photographic evidence from the observed landscape and compare them to cross-sections from the simulated results.

[Figure]

Figure 3. (a) The simulation discharge in 2011-2013 in Scenario PP; (b) the verification location; (c) the comparison of the simulated to the recorded discharge.

[Figure]

(a) the field measurement location

(b) simulated elevation changes after 2013 in Scenario PP

(c) photos after September 2013

Figure 4. The comparison of cross-sections from the simulation results to the photos in the field measurement locations after 2013 in Scenario PP.

I would recommend that the authors include in the discussion notes on how the outputs of this analysis could be used – ie, why is this work useful. Is the intention that these modelling approaches will be used in the future to design debris-flow management schemes and help to inform decision making, for example?

**6. The discussion of application**

Followed by the recommendations, we enrich the first section in discussion to model uncertainty and application. The application was discussed as follows.

The methods applied in the study further demonstrate the role of C-L as a tool to understand the short-medium term or the long-term geomorphology changes (Ramirez et al., 2022; Li et al., 2020; Coulthard et al., 2012) and observe the effectiveness of natural hazard interventions measures provided different rainfall patterns . For example, the mitigation facilities in this study were effective, especially engineering measures

that cooperated with vegetation revetments in the upstream area, which would help decision-makers to optimize the management strategies to control mountain disasters. Geotechnical engineering has its disadvantages event though it is a mature technology that identifies and fixes problems quickly (Peng and Yongming, 2013), such as the greater work and expense and the difficulty of maintenance. While the 'green development', the vegetation cover was effective to prevent erosion by strengthening topsoil and absorbing excess rainwater with its roots (Reichenbach et al., 2014; Stokes et al., 2014; Forbes and Broadhead, 2013; Mickovski et al., 2007). Alternatively, the methods could be used to study the tree planting patterns in different slopes.

On the language in the manuscript, I found the vast majority of the manuscript well written and easy to follow. There were a few instances where phrasing is not quite comfortable and I think some editorial guidance would be sufficient to improve these. Some in-line references contain initials and these should be corrected.

Thanks for the suggestions and we would polish the language and revise the references in the new manuscript.

**Reference**

Coulthard, T. J. and Skinner, C. J.: The sensitivity of landscape evolution models to spatial and temporal rainfall resolution, Earth Surf. Dyn., 4, 757–771, https://doi.org/10.5194/esurf-4-757-2016, 2016.

Coulthard, T. J., Hancock, G. R., and Lowry, J. B. C.: Modelling soil erosion with a downscaled landscape evolution model, Earth Surf. Process. Landforms, 37, 1046–1055, https://doi.org/10.1002/esp.3226, 2012.

Feng, W., He, S., Liu, Z., Yi, X., and Bai, H.: Features of Debris Flows and Their Engineering Control Effects at Xinping Gully of Pingwu County, J. Eng. Geol., 25, 2017.

Forbes, K. and Broadhead, J.: Forests and landslides: the role of trees and forests in the prevention of landslides and rehabilitation of landslide-affected areas in Asia, FAO, 14–18 pp., 2013.

Goldberg, D. E.: Genetic Algorithms in Search, Optimization and Machine Learning, 1st ed., Addison-Wesley Longman Publishing Co., Inc., USA, 1989.

Guo, Q., Xiao, J., and Guan, X.: The characteristics of debris flow activities and its optimal timing for the control in Shikan River Basin Pingwu Country, Chinese J. Geol. Hazard Control, 29, 2018.

Lee, T. and Jeong, C.: Nonparametric statistical temporal downscaling of daily precipitation to hourly precipitation and implications for climate change scenarios, J. Hydrol., 510, 182–196, https://doi.org/10.1016/j.jhydrol.2013.12.027, 2014.

Li, C., Wang, M., Liu, K., and Coulthard, T. J.: Landscape evolution of the Wenchuan earthquake-stricken area in response to future climate change, J. Hydrol., 590, 125244, https://doi.org/10.1016/j.jhydrol.2020.125244, 2020.

Mickovski, S. B., Bengough, A. G., Bransby, M. F., Davies, M. C. R., Hallett, P. D., and Sonnenberg, R.: Material stiffness, branching pattern and soil matric potential affect the pullout resistance of model root systems, Eur. J. Soil Sci., 58, 1471–1481, https://doi.org/10.1111/j.1365-2389.2007.00953.x, 2007.

Peng, C. and Yongming, L.: Debris-Flow Treatment: The Integration of Botanical and Geotechnical Methods, J. Resour. Ecol., 4, 097–104, https://doi.org/10.5814/j.issn.1674-764x.2013.02.001, 2013.

Ramirez, J. A., Mertin, M., Peleg, N., Horton, P., Skinner, C., Zimmermann, M., and Keiler, M.: Modelling the long-term geomorphic response to check dam failures in an alpine channel with CAESAR-Lisflood, Int. J. Sediment Res., 37, 687–700, https://doi.org/10.1016/j.ijsrc.2022.04.005, 2022.

Reichenbach, P., Busca, C., Mondini, A. C., and Rossi, M.: The Influence of Land Use Change on Landslide Susceptibility Zonation: The Briga Catchment Test Site (Messina, Italy), Environ. Manage., 54, 1372–1384, https://doi.org/10.1007/s00267-014-0357-0, 2014.

Stokes, A., Douglas, G. B., Fourcaud, T., Giadrossich, F., Gillies, C., Hubble, T., Kim, J. H., Loades, K. W., Mao, Z., McIvor, I. R., Mickovski, S. B., Mitchell, S.,

Osman, N., Phillips, C., Poesen, J., Polster, D., Preti, F., Raymond, P., Rey, F., Schwarz, M., and Walker, L. R.: Ecological mitigation of hillslope instability: Ten key issues facing researchers and practitioners, Plant Soil, 377, 1–23, https://doi.org/10.1007/s11104-014-2044-6, 2014.

---

## Author Response (AR1)

**Point-by-point replies to the referees' comments**

Title: "Assessment of Short-medium Term Intervention Effects Using CAE2 SAR-Lisflood in Post-earthquake Mountainous Area"

Authors: Di Wang, Ming Wang, Kai Liu

Manuscript Number: nhess-2022-195

We thank all the referees for reviewing our manuscript and making highly constructive comments and suggestions. The comments and suggestions were formatted in light and dark blue text. The author's response to the major issues is shown below in black text.

**Referee Jorge Ramirez:**

**General comments**

In this study, a landscape evolution model, CAESAR-Lisflood (CL), is applied to a steep mountain catchment to assess the effectiveness of engineering works in reducing the transport of sediment. This is an applied study, that is straightforward, and demonstrates the use of CL in a highly dynamic landscape. Overall, the manuscript fits the scope of NHESS and would be interesting to modellers and practitioners working in mitigating geo hazards in mountainous regions. My concerns with the study are related to the choice of hydrological parameters, the physical plausibility of landscape changes, and the development of initial conditions. In addition, the clarity of the manuscript requires
substantial improvement and I recommend the text is thoroughly edited by a native English speaker, with a background in fluvial geomorphology, before acceptance. Below are major comments that need to be addressed followed by a list of minor points and edits.

**Minor points and edits:**

In this part, we corrected, changed and rewrote the information and sentences according to the minor suggestions directly in the manuscript. Here we answer the questions.

Table 1: "mess of farmland" needs to be reworded. Maps in table are quite small, perhaps remove them from the table and make them larger in a multi panel supplemental information figure, and reference this in the text. In the table you could add the summary statistics of the landslides derived from the maps (e.g. total area, min area, max area).

Thanks for the suggestions. We removed Table 1 from the manuscript and reported it in the Supplemental Material to keep the manuscript concise. And we decided to delete the landslide scar distribution from the remotely sensed images and remove all the relative information from the table.

Line 130: what do you mean by "lower reservoir was full of loose material.", be more specific.

We rewrote another sentence: "Nearly three years later, the storage capacity behind the upper dam remained 50% in 2016, while the transitional area dam cannot retain sediment.".

Line 145-146: Provide an example of a vegetation revetment.

We referenced Lan et al. (2020) and took the trees planting as an example.

Line 152: Change to NHESS citation format: (Tom J Coulthard et al., 2013)

Line 152: Change to NHESS citation format: (T J Coulthard et al.)

We confirmed that we cited articles formatted in the Copernicus publications, which NHESS asks for. And in-text citations linked the last name with the publication year between brackets.

Line 185: Explicitly mention if the UP DEM does or does not contain dams.

We changed the "DEM" into "the non-sinks DEM" to underline there were no additional operations.

Line 192: You mention a field survey and how this led you to conclude the thickness of erodible sediment was spatially variable. In the text, elaborate more about the field survey and how it informed the production of the bedrock layers in CL.

First, we replaced the complex start with "In terms of Section 2.2 (Herein, the source materials were introduced)". And then, we showed the way of producing the approximate thickness values, estimated by the relative elevation of various objects to the ground level in five regions.

Line 205: define bedDEMs and "In addition, DEMs were formatted to ASCII raster as required by C-L"

We defined the bedDEMs in the first sentence of this paragraph.

Fig 2.: The flowchart is redundant as the text has already described the process quite well. I like the figure, but without the flowchart and then change the caption accordingly.

Considering the locations of the additional levees and vegetation revetments introduced in Section 3.2.2 that we applied in Scenario EP in this figure, we removed it from the manuscript and added in the supplement material. And the captions were changed accordingly.

Table 2: Explain the choice of setting the slope failure th "eshold so high (60 deg), wouldn't this prevent the occurrence of landslides and lead to deep canyonlike incision?

Considering the steep slope on both sides of deep gullies there distribute, we tended to set a higher slope failure threshold to replicate the geomorphic changes realistically. Additionally, we found that the probability of shallow landslides indeed accumulates from 20°to 50° in slope gradient between 2011 and 2013 (Li et al., 2020). The slope angle was derivate from the DEM with 30 m spatial resolution, which caused a lower slope angle than that with a 10 m resolution. As such, we set 60°, which is lower than the 65° used in a scenario without landslides (Xie et al., 2022b) and higher than the 50°.

Line 269-270: Here you mention "damage" but what you are really calculating is exposure because you are not calculating a monetary value. Change the text to mention exposure, and I am assuming that a map of settlements or landcover was used to calculate the exposure, if so, provide this map in the supplemental information. If you didn't use a landcover map, explain what you used to derive exposure.

The "damage" used here might be misleading. We rewrote it into "The total area of affected grid cells representing erosion and deposition in each scenario was calculated to reveal the difference obviously. And the areas varied in depths were used to compare the extent of geomorphic changes in three situations.".

Line 317-327: The narrative you provide is fine, but you need to provide some values that support your comparisons.

We rewrote the paragraph and underlined that "the areas of extreme, moderate, and light deposition decreased in the order of UP, PP, and EP, and the heavy deposition areas show the opposite trend, which mainly contributes to the checking dams and vegetation revetments."

Line 341-351: Add values to support the existing qualitative comparison.

We qualified using Fig.6 and Fig. 7 to add the comparative results.

In Figure 6, the EP map shows the levees blocking a tributary, is this a mistake in the figure or did you really block this tributary in the simulation. Please explain.

We have checked carefully and ensured that the levees did not block the deep and narrow outlet of the

tributary.

Line 372: "Divisional erosion and deposition" doesn't make sense

We removed this headline and combined it with the headline of relative effectiveness.

There introduced descriptions of the similarities. We provided the qualitative comparison in the next paragraph.

Figure 8: I am unsure what is plotted here or what type of plot this is. Is this the distribution of simulated geomorphic change after three years in the source, translational, and deposit areas? If so, are these boxplots? If these are boxplots, provide the median, IQR, whiskers and outliers. Add more explanation to the figure caption and the y-axis label should read "    ".

We showed a new figure to explain better.

Conclusion section: Here you need to further summarize your main findings because the current text reads like a repetition of the results.

We have summarized four findings in a paragraph.

**Major comments**

A weakness of the study is the lack of **calibration of the hydrological component in CL**. As such, there is no way of knowing if the quantity and timing of the floods in the ungauged catchment are accurately replicated by CL. The hydrological parameters adopted (m-values) are from studies performed from nearby catchments but these studies also have not performed calibration to derive m-values. The authors, instead rely on landcover to assign m-values, but **m-value is only partly dependent on landcover**. For example, Ramirez et al.2022 found that in a mountain catcInstitute'sdeESRI'**srrelated well with m-value** and not with landcover. To have greater confidence in the model, the authors need to provide **hydrographs for the entire simulated period** and, in addition, provide qualitative or quantitative data that confirms the physical plausibility of the simulated discharge, specifically the floods.

1.    **The m-values**

The m-values in C-L influence the peak and duration of the hydrograph in response to rainfall (Coulthard et al., 2002), which are usually determined by the landcover (e.g., 0.02 for the forest, 0.005 for the grassland) (Coulthard and Van De Wiel, 2017). In our study, we united the value to 0.008 in our smaller catchment (14 km$^2$) in Scenario UP and PP, which resembles the m-value of farmland covered with lower vegetation in the same catchment studied by Xie et al. (2022), Li et al., (2020) and Xie et al., (2018). In scenario EP, the m-value in the vegetation revetments area was 0.02 to distinguish the vegetation coverage. It has been calibrated in the more extensive catchment containing our study area (Xie et al., 2018) by replicating the flood event in 2013.

We have read Ramirez et al. (2022) intensively and learned the new and creative method to calibrate components. They determined m-values in the total catchment after perfect simulation results in a sub-catchment according to the soil depth. And we think

It would exist discrepancies between different regions. In our catchment, the vegetation counts more than the soil depth for the m-value, which is caused by the undetermined soil depth. Therefore, it is not the optimal method for our smaller study area distributed larger amount of landslides deposition and river alluvium stem from metamorphic sandstones and sandy slate.

**2. Calibration**

Admittedly, it is not enough for our calibration work to reference the parameters from the published research in the same catchment and use the recommended values by model developers. Considering the ungauged basins before 2015, we replicated the flash flood event in July 2018 by C-L to calibrate the hydrological components.

There are no considerable differences in geomorphology, channel location, and land cover before 2013 and after 2018 in our catchment found from the field surveys. Based on Scenario PP (with two check dams), we changed the rainfall series to the two-week hourly precipitation in July 2018, which is recorded by the rain gauge 2.5 km away from the catchment placed in 2015. The simulation results (Figure 1c and Figure 1d) showed the erosion and maximum flood depth deposition distributions in Scenario PP on July 15th, 2018. As shown in Figure 1c and Figure 1d, we selected three locations randomly to compare the simulation results with remotely sensed images and photos. The results (Figure 2) showed reliable results, including sediment deposition and the peak flood depth, which indicate that the flash flood event was replicated successfully by the C-L.

[Figure]

**Figure 1: The input rainfall series (a and b) and simulation results of the flash flood event in July 2018 (c and d).**

[Figure]

| | (a) modeling results | (b) images in June 2019 (after flooding) | (c) photos in September 2018 (after flooding) |

**Figure 2: The comparison of the simulation results with images (GF-2 with 8-m resolution) and photographic evidence after the flash flood event in July 2018.**

**3. Verification**

In section 5.1, we confirmed the plausibility of the simulated results using deposition depth evaluated from photos. Herein, we add the discharge of the entire simulation period for Scenario PP. As shown in Figure 3, we compare two types of discharge recorded in published research (Feng et al., 2017; Guo et al., 2018) with those of simulation results to confirm the physical plausibility. And we captured five flood events where the daily precipitation was more than 50 mm in 2013, and the peak discharge was up to 63.6 m³ /s.

[Figure]

**Figure 3: (a) The simulation discharge in 2011-2013 in Scenario PP; (b) the verification location; (c) the comparison of the simulated to the recorded discharge.**

In this study, CL simulations have produced locations of deep erosion between 3-10 m in a period of three years. This is quite a bit of erosion in such a short period and in some instances would produce features in the simulated landscape that resemble small canyons. Could you **verify that these erosional features are physically plausible** by providing photographic evidence from the observed landscape and comparing them to cross-sections from the simulation? Or provide any other type of validation that

supports such extreme erosion across the simulated landscape. In addition, across all simulations (Fig. 5a), there are instances of erosion that exceed 3 m in the downstream area where erodible thickness is 3 m. Can the authors explain how **simulated erosion can exceed the thickness of the initial erodible sediment?** Likewise, in this study, how is it possible for CL to produce erosion between 10 and 15 m, if the maximum depth of erodible sediment in the catchment is 10 m?

**4.    Revised erosional and deposited features**

Many thanks for your reminder, and we apologize for our mistakes in showing the abandoned simulation result, where the input basedDEMs were generated improperly. Now we update the revised figures as shown in Figure 4 and Figure 5, showing the spatial distribution of erosion and deposition. We correct the description (extreme erosion (<-7 m), heavy erosion (-7--3 m), moderate erosion (-3--1 m), light erosion (-1-0.1 m), micro change (-0.1-0.1 m), light deposition (0.1-1 m), moderate deposition (1-3 m), heavy deposition (3-7 m), and extreme deposition (>7 m)) in section 4.1 and 4.2. After checking all the figures, tables and numbers, we ensure that all the analysis results and input parameters are consistent and from our optimal simulation.

In addition, we thank the reviewer for his suggestions about the figure's details.

[Figure]

**Figure 4: (a) Simulated geomorphic changes over time for three scenarios; (b) the exposure area included deposition and erosion for three scenarios; (c) the distribution of deposition and erosion at the conclusion of the simulation for the three scenarios.**

[Figure]

**Figure 5: Geomorphic changes at the conclusion of the simulation at key spots for the UP, PP, and EP scenarios. The top row is the upriver section containing dam 1, dam 2 and the vegetation revetment. The bottom row is the downriver section containing levees.**

**5. Verification of erosion and deposition**

As shown in Figure 6, we verify the erosional and deposited features by providing photographic evidence from the observed landscape and compare them to cross-sections from the simulated results.

[Figure]

(a) the field measurement location

(b) simulated elevation changes after 2013 in Scenario PP

(c) photos after September 2013

**Figure 6: The comparison of cross-sections from the simulation results to the photos in the field measurement locations after 2013 in Scenario PP.**

In the study, there is no mention of establishing initial conditions by spinning-up the model to mix the grain sizes. **If spin-up was not performed, can the authors provide an explanation.** If spin-up was performed, could you briefly explain how it was done in the methods. Regarding the choice of bedrock elevation (Fig. 2), could the authors provide the physical basis for the choice of erodible thickness values and locations of these values.

**6.    Spin-up processes**

Admittedly, we didn't spin up the model to mix the grain sizes. The purposes of the process are to eliminate the 'walls' and the 'depressions' in the cells and avoid the intense erosion in the hill slope in the early run time. Actually, we preprocess the DEMs by filling the sink based on Environmental Systems Research Institute's (ESRI's) ArcMap (ArcGIS, 10.8) to eliminate the problematic pixels. Moreover, for our catchment, the fine grains were distributed homogeneously in the hill slope and the channel five years after the strong earthquake. Therefore, we think a huge difference would not exist. However, we will continue to compare the difference in future work.

**7.    Erodible thickness values**

It is difficult to provide direct evidence like drilling operations and the fine map because of the steep terrain and a large amount of landslides deposition in these post-earthquake fragile mountains. The bedrock elevation was evaluated mainly from the published research from the trusted official institutions in China (Feng et al., 2017; Guo et al., 2018) and verified with our field survey. The institutions described their research using the prompt and accurate hazard inventory and the UAV survey.

**Referee #2**

Here we added headlines to those comments that required more details

First and foremost, is **the lack of proficiency and fluency in the use of the English language and grammar**, which is consistently poor throughout this paper. This makes it very difficult to comprehend the contents of the paper – for example, it is not clear what the methods were, or how sets of parameters used in simulations were obtained/derived. Because of this, it was also not clear how the results were obtained and what they actually represented, and consequently, whether the resulting discussion and conclusions could be substantiated or supported. Overall, while the aims and objectives could be understood, it was not easy to determine if they had been met. Unfortunately, the author's unfamiliarity with the English language meant that too many sentences were variously incomplete, made no sense, or utilized inappropriate or misspelt words.

Admittedly, the English language in this research is not proficient and fluent. We have revised carefully in the manuscript.

Some comments and suggestions for improvement:

Rather than try to describe the background to the CAESAR model and how it works themselves, I believe the authors could more clearly and succinctly acknowledge this by referencing existing publications which describe this ie. Coulthard et al 2012.

As C-L users, we introduced the background of the C-L and the relevant parameters' definitions to understand well, and we rewrote this section concisely in the manuscript.

As currently written, descriptions of specific parameters and methods used in this study and the scenarios modelled are poorly described, or not described at all – for example:

a table of parameters lists values used in simulations, yet **there is no explanation of how or why the values in the table were selected or used** – for example, those representing vegetation parameters (shear stress, age to maturity, proportion of erosion) - were selected and utilized in model simulations for the scenarios in this study.

**8.    Parameters**

We admit that some parameters were not introduced in detail, except for the definitions, such as shear stress, age to maturity, and proportion of erosion.

Some sensitive parameters to the model (Skinner et al., 2018) were referenced from some published research in the same catchment, such as the sediment transport formula, grain size and corresponding proportion and the slope failure threshold (Xie et al., 2018; Li et al., 2020; Xie et al., 2022a). In addition, we add the calibration of the components in C-L in the first reply. Other parameters in Table 2 are from the default values recommended by the developers (such as the max erode limit in the erosion/deposition module and the vegetation critical shear stress) in https://sourceforge.net/p/caesar-lisflood/wiki/Home/.

The authors **do not explain why they selected some of the parameters ie why a specific sediment transport equation** was selected. Depending on the sediment transport equation applied, very different model results may occur.

The sediment transport equation was sensitive to the C-L model (Skinner et al., 2018), and we selected the Wilcock and Crowe referenced from published research studied in the same catchment (Xie et al., 2018; Li et al., 2020; Xie et al., 2022a). The Wilcock-Crowe equations are among the more widely used formulae for predicting fractional bed load transport rates in gravel bed streams. They are developed using bed load transport information obtained in laboratory flume experiments with bed material

sediments ranging in particle size from 2.83 to 64 mm (Wilcock et al., 2003). Another alternative equation is Einstein-Brown, developed by uniform sediment and lightweight materials ranging in sizes from 0.785 to 28.65 mm based on flume data. From the grain sizes in our study area, the Wilcock-Crowe equations would be the better choice.

It is not clear **what rainfall data was used in the simulations** – whether different sets of data were used for different scenarios, or one set was applied across all scenarios. The text about the downscaling rainfall data is simply confusing and does not address this.

**9.    The downscaled rainfall data**

Firstly, we revised the description of the temporal downscaling process as follows.

In this research, we compared three scenarios using identical precipitation data during 2011 and 2013, as mentioned in section 3.1. The daily source precipitation of one station in 2011-2013 was downloaded from China Meteorological Administration (http://data.cma.cn). The rainfall intensity and the frequency of extreme events affect patterns of erosion and deposition (Coulthard and Skinner, 2016; Coulthard et al., 2012). And we used the stochastic downscaling method to generate hourly data to best capture the hydrological events in this study, which was introduced by (Li et al., 2020; Lee and Jeong, 2014). The referenced hourly precipitation was measured from the pluviometer located 20 km from the study area in 2016, with annual total precipitation of 684 mm. The rainfall in 2016 was characterized by (1) hourly precipitation from 1.1 mm to 35.4 mm and (2) the maximum and average duration of a rainfall event up to 24 h and 2.8 h. The main processes of the downscaling method are as follows.

- extracting the measured daily rainfall closest to the referenced daily rain in 2011-2013 through the threshold setting and producing the genetic operators from the extracted hourly precipitation;
- mixing on the genetic operators by genetic algorithm (Goldberg, 1989) composed of reproduction, crossover and mutation and repeating until the distance between the predicted daily rainfall and the measured rainfall is less than the threshold;
- normalizing the hourly precipitation to remain the daily rainfall value unchanged.

The input of generated hourly precipitation is catchment lumped in Scenario UP and EP and divided into two separate but identical rainfall in Scenario EP.

Additionally, we added Table 1 to address the scenarios settings and the input rainfall in the revised manuscript.

**Table 1: The main settings and input for the three scenarios.**

| Scenario | Descriptions | Period | DEM (10 m) | Rainfall data |
|---|---|---|---|---|
| UP | no anthropogenic intervention | | UP DEM UP bedDEM | downscaled hourly precipitation in the period (lumped) |
| PP | the present two blocking dams upstream without dredging work | 2011-2013 (3 years) | PP DEM PP bedDEM | |
| EP | additional vegetation revetments in the source area and levees in the deposit area based on Scenario PP | | EP DEM EP bedDEM | downscaled hourly precipitation in the period (spilt) |

As written, it is not clear how results are obtained or substantiated from the methods. The authors make assumptions about the ability of the model to erode that are not supported or substantiated by any evidence. Specifically, the authors describe how they have attempted to incorporate levees and dams into simulations by simply increasing the elevation in certain areas and not changing other parameters such as particle size. While this reviewer agrees it may temporarily reduce flow, in the longer term, this may

well lead to increased erosion in other areas around the sides of the dam / levee.

**10.  Verification and assumptions**

We have added the verification in the first reply section 3 and 5, including comparing the simulation results with the hydrographs and photographic evidence.

We incorporated levees and dams into simulations by changing the surface DEMs and bedDEMs of dams and levees described in section 3.2.1.

Admittedly, there were many assumptions in our simulation work, and we ignored the particle size changes before and after the dams and levees. The study of increased erosion around the sides of the dams and levees is limited in the C-L model because of the unattackable settings near the engineering facilities. Actually, we paid more attention to the short-medium effectiveness of the interventions, which represented the regional features analyzed from the erosion and deposition in the total catchment and the output in the outlet. If possible, we would study from a smaller scale by using other simulation models or field surveys in future work.

The authors have demonstrated **a poor use of figures and tables** to support their results. Specifically, figures variously lack scales or annotations to indicate where the areas of erosion or deposition are (eg figure 5), or where other features (such as dams) referenced in the text are located (eg figure 1).

Thanks for the suggestions. We added the scales before, but they looked odd because the figure aims to show the distribution of erosion and deposition. At last, we decided to remove the scales and other annotations in the newest one.

Some figure captions do not make sense eg figure 2 does not clearly show any chart or process for generating the bedrock DEMs. Some figures do not appear to contain the information described in the text.

Thanks for the suggestions. We removed this figure from the manuscript.

Tables are present in the manuscript which are not referenced in the text; different tables share the same number (eg there are 2 tables labelled as table 2); and some tables do not identify what the units in the table represent.

We checked again and revised the manuscript.

Finally, the author's use of referencing is poor and inconsistent.

Thanks for the suggestions on references. We have revised the references using the standard styles.

**Referee Christopher Skinner:**

In the second Table 2 (there are two) on page 22, the sediment conservation ability in the source area increases from 0.5 to 0.55 between the PP and EP scenarios, yet I don't think there are any additional interventions in this area (levees and vegetation are in the deposit area). I think this may be an effect of the way the authors have applied the spatially varied "m" in the model. In UP and PP, the "m" has a global value of 0.008 and in EP the vegetated areas are given a separate value of 0.02. It isn't stated by the authors but I believe **their rainfall input is catchment lumped (please could the authors confirm)**. My understanding of CL is that for a lumped input it will average all the "m" values and create a single lumped input from it, in this case making the input for the whole catchment less flashy. Alternatively, the authors could **specify two separate rainfall input areas**, one for the vegetated area and one for the rest of the catchment, in effect making two hydrological response units (HRUs) for the model, each with its own input based on the local "m" value. I don't think this needs to be done for a revised manuscript as I doubt it would change their conclusions materially, but it should at least be acknowledged.

Firstly, we have corrected the tables' labels. We agree the increase of the sediment conservation ability between the PP and EP scenarios is an effect of the spatially varied m values in the model, which supported the stabilizing sediment with vegetation roots.

Further on "m", I concur with the comments from Jorge that where possible the value should be calibrated against gauged data. If this is not available, basing the value on land cover, as the authors have done, is reasonable. However, the authors are using downscaled hourly rainfall, not observed hourly rainfall, so **any calibration would need to account for this**.

I also concur with Jorge's comment on **spin-up period**. There are no details in the manuscript and it would be helpful to know. In this case, where much of the eroded material is fresh and loose, it could be argued a spin-up might actually be a counterproductive in this instance.

**11. Calibration and spin-up processes**

We answered the same questions in section 2 and 6, where we replied to referee Jorge Ramirez.

The downscaling of the rainfall to hourly is really important (as shown nicely in Figure 3). Sorry to push one of my papers, but Coulthard and Skinner (2016: https://doi.org/10.5194/esurf-4-757-2016) provides some analysis of why and it would be useful to refer to this here. Unfortunately, I found the description of how this was done not clear – please could the authors **revisit this description** so it is easier to follow. It would be useful to also know the spatial resolution of the rainfall product used and the spatial resolution it applied to the model with (I assumed it was lumped).

**12. Rainfall data**

Thanks for the recommendation. The findings from a range of simulations have revealed that using time-averaged climate inputs may be under-predicting basin sediment yields and introducing spatial biases through under-predicting or over-predicting erosion (Coulthard and Skinner, 2016). We have referenced in our revised manuscript to explain why we downscale the daily rainfall into hourly rainfall.

The description of the downscaling method was revised in section 9. The input of generated hourly precipitation is catchment lumped in Scenario UP and PP and divided into two separate but identical rainfall in Scenario EP. We added a table described in section 9.

A **verification of the model outputs** for the PP scenario by comparing them to real-world observations would strengthen the analysis of the paper. For example, Figure 10 and related discussion could be

included within the results as a form of verification for the model outputs.

**13. Verification**

Similar to the reply in section 3 and 5, we add the verification in the discharge and erosion/deposition at crucial spots. As shown in Figure 3, we compare two types of discharge recorded in published research (Feng et al., 2017; Guo et al., 2018) with those of simulation results to confirm the physical plausibility. As shown in Figure 4, we verify the erosional and deposited features by providing photographic evidence from the observed landscape and compare them with cross-sections from the simulated results.

I would recommend that the authors include in the discussion notes on how the outputs of this analysis could be used – ie, why is this work useful. Is the intention that these modelling approaches will be used in the future to design debris-flow management schemes and help to inform decision making, for example?

**14. The discussion of the application**

Followed by the recommendations, we enrich the first section in the discussion about model uncertainty and application. The application was discussed as follows.

The methods applied in the study further demonstrate the role of C-L as a tool to understand the short-medium term or long-term geomorphology changes (Ramirez et al., 2022; Li et al., 2020; Coulthard et al., 2012) and observe the effectiveness of natural hazard interventions measures provided different rainfall patterns. For example, the mitigation facilities in this study were effective, especially engineering efforts that cooperated with vegetation revetments in the upstream area, which would help decision-makers to optimize the management strategies to control mountain disasters. Geotechnical engineering has disadvantages, even though it is a mature technology that identifies and fixes problems quickly (Cui and Lin, 2013), such as the greater work and expense and the difficulty of maintenance. While the 'green development', the vegetation cover was effective to prevent erosion by strengthening topsoil and absorbing excess rainwater with its roots (Reichenbach et al., 2014; Stokes et al., 2014; Forbes and Broadhead, 2013; Mickovski et al., 2007). Alternatively, the methods could be used to study the tree planting patterns on different slopes.

On the language in the manuscript, I found the vast majority of the manuscript well written and easy to follow. There were a few instances where phrasing is not quite comfortable and I think some editorial guidance would be sufficient to improve these. Some in-line references contain initials and these should be corrected.

Thanks for the suggestions and we have polished the language and revised the references in the new manuscript.

**Reference**

Coulthard, T. J. and Skinner, C. J.: The sensitivity of landscape evolution models to spatial and temporal rainfall resolution, Earth Surf. Dyn., 4, 757–771, https://doi.org/10.5194/esurf-4-757-2016, 2016.

Coulthard, T. J. and Van De Wiel, M. J.: Modelling long term basin scale sediment connectivity, driven by spatial land use changes, Geomorphology, 277, 265–281, https://doi.org/10.1016/j.geomorph.2016.05.027, 2017.

Coulthard, T. J., Macklin, M. G., and Kirkby, M. J.: A cellular model of Holocene upland river basin and alluvial fan evolution, Earth Surf. Process. Landforms, 27, 269–288, https://doi.org/10.1002/esp.318, 2002.

Coulthard, T. J., Hancock, G. R., and Lowry, J. B. C.: Modelling soil erosion with a downscaled landscape evolution model, Earth Surf. Process. Landforms, 37, 1046–1055, https://doi.org/10.1002/esp.3226, 2012.

Cui, P. and Lin, Y.: Debris-Flow Treatment: The Integration of Botanical and Geotechnical Methods, J. Resour. Ecol., 4, 097–104, https://doi.org/10.5814/j.issn.1674-764x.2013.02.001, 2013.

Feng, W., He, S., Liu, Z., Yi, X., and Bai, H.: Features of Debris Flows and Their Engineering Control Effects at Xinping Gully of Pingwu County, J. Eng. Geol., 25, 2017.

Forbes, K. and Broadhead, J.: Forests and landslides: the role of trees and forests in the prevention of landslides and rehabilitation of landslide-affected areas in Asia, FAO, 14–18 pp., 2013.

Goldberg, D. E.: Genetic Algorithms in Search, Optimization and Machine Learning, 1st ed., Addison-Wesley Longman Publishing Co., Inc., USA, 1989.

Guo, Q., Xiao, J., and Guan, X.: The characteristics of debris flow activities and its optimal timing for the control in Shikan River Basin Pingwu Country, Chinese J. Geol. Hazard Control, 29, 2018.

Lan, H., Wang, D., He, S., Fang, Y., Chen, W., Zhao, P., and Qi, Y.: Experimental study on the effects of tree planting on slope stability, Landslides, 17, 1021–1035, https://doi.org/10.1007/s10346-020-01348-z, 2020.

Lee, T. and Jeong, C.: Nonparametric statistical temporal downscaling of daily precipitation to hourly precipitation and implications for climate change scenarios, J. Hydrol., 510, 182–196, https://doi.org/10.1016/j.jhydrol.2013.12.027, 2014.

Li, C., Wang, M., Liu, K., and Coulthard, T. J.: Landscape evolution of the Wenchuan earthquake-stricken area in response to future climate change, J. Hydrol., 590, 125244, https://doi.org/10.1016/j.jhydrol.2020.125244, 2020.

Mickovski, S. B., Bengough, A. G., Bransby, M. F., Davies, M. C. R., Hallett, P. D., and Sonnenberg, R.: Material stiffness, branching pattern and soil matric potential affect the pullout resistance of model root systems, Eur. J. Soil Sci., 58, 1471–1481, https://doi.org/10.1111/j.1365-2389.2007.00953.x, 2007.

Ramirez, J. A., Mertin, M., Peleg, N., Horton, P., Skinner, C., Zimmermann, M., and Keiler, M.: Modelling the long-term geomorphic response to check dam failures in an alpine channel with CAESAR-Lisflood, Int. J. Sediment Res., 37, 687–700, https://doi.org/10.1016/j.ijsrc.2022.04.005, 2022.

Reichenbach, P., Busca, C., Mondini, A. C., and Rossi, M.: The Influence of Land Use Change on Landslide Susceptibility Zonation: The Briga Catchment Test Site (Messina, Italy), Environ. Manage., 54, 1372–1384, https://doi.org/10.1007/s00267-014-0357-0, 2014.

Skinner, C. J., Coulthard, T. J., Schwanghart, W., Van De Wiel, M. J., and Hancock, G.: Global

sensitivity analysis of parameter uncertainty in landscape evolution models, Geosci. Model Dev., 11, 4873–4888, https://doi.org/10.5194/gmd-11-4873-2018, 2018.

Stokes, A., Douglas, G. B., Fourcaud, T., Giadrossich, F., Gillies, C., Hubble, T., Kim, J. H., Loades, K. W., Mao, Z., McIvor, I. R., Mickovski, S. B., Mitchell, S., Osman, N., Phillips, C., Poesen, J., Polster, D., Preti, F., Raymond, P., Rey, F., Schwarz, M., and Walker, L. R.: Ecological mitigation of hillslope instability: Ten key issues facing researchers and practitioners, Plant Soil, 377, 1–23, https://doi.org/10.1007/s11104-014-2044-6, 2014.

Wilcock, P. R., Asce, M., and Crowe, J. C.: Surface-based Transport Model for Mixed-Size Sediment Surface-based Transport Model for Mixed-Size Sediment, 9429, https://doi.org/10.1061/(ASCE)0733-9429(2003)129, 2003.

Xie, J., Wang, M., Liu, K., and Coulthard, T. J.: Modeling sediment movement and channel response to rainfall variability after a major earthquake, Geomorphology, 320, 18–32, https://doi.org/10.1016/j.geomorph.2018.07.022, 2018.

Xie, J., Coulthard, T. J., and McLelland, S. J.: Modelling the impact of seismic triggered landslide location on basin sediment yield, dynamics and connectivity, Geomorphology, 398, 108029, https://doi.org/10.1016/j.geomorph.2021.108029, 2022a.

Xie, J., Coulthard, T. J., Wang, M., and Wu, J.: Tracing seismic landslide-derived sediment dynamics in response to climate change, Catena, 217, 106495, https://doi.org/10.1016/j.catena.2022.106495, 2022b.

---

## Author Response (AR2)

**Point-by-point replies to the referees' comments**

Title: "Assessment of Short-medium Term Intervention Effects Using CAE2 SAR-Lisflood in Post-earthquake Mountainous Area"

Authors: Di Wang, Ming Wang, Kai Liu, Jun Xie

Manuscript Number: nhess-2022-195

We thank the co-editor and referees #1 Jorge Ramire for reviewing our revised manuscript and the editor for giving an opportunity to improve it. The suggestions were formatted in light blue text. The author's response is shown below in black text.

I thank the authors for revising the manuscript and making it considerably better than the previous version. Although much improved, the text needs further refinement and mistakes occur repeatedly throughout the entire manuscript. I highly recommend that the text is further polished before NHESS accepts this for publication.
Figure 1: Yellow text in legend is not legible and gray dashed line is barely visible on map.

We replaced the yellow text with dark red ones and changed the gray dashed line to white dashed line in Fig.1, which has high colour contrast and looks better.

[Figure]

**Figure 1: Overview of the study area. (a) Location of study area; (b) Seismic intensity map of the Wenchuan earthquake within the Pingwu county; (c) The schematic image of the study area.**

Line 122: How does Figure 2 support this sentence?

We mistook the figure and the location of the parenthesis. The correct sentence is "Considering the damages flash-flood caused to the residential area downstream, the levees (see Fig. S1 and Section 3.2.2) are artificial barriers to protect agricultural land and buildings".

Figure 2b: Caption and plot title indicate year 2016, but the plot axis is labelled 2011.

We corrected the axis labels shown in Fig. 2.

[Figure]

**Figure 2: (a) Daily precipitation in 2011-2013 (the red vertical line indicates maximum daily precipitation of 126.5 mm); (b) Hourly precipitation in 2016; (c) Downscaled hourly precipitation in 2011-2013 (the red horizontal line indicates the hourly-mean precipitation 5.27 mm on the day with maximum precipitation marked in (a)).**

Figure S3: In the text, explain how you obtained the scale on these images?

We explained as "Figure S3: The comparison of the simulation results (labelled with a depth range of deposition and inundation in the delimited regions shown in (b)) with images (GF-2 with an 8-m resolution, annotated three locations photographed in (c)) and photographic evidence (dimensioned to show the measured results) after the flash flood event in July 2018" in the title.

Figure 5a needs a legend and scale bar.

We added a legend and scale bar shown in Fig. 3.

[Figure]

(a) the field measurement location

(b) simulated elevation changes after 2013 in Scenario PP

(c) photos after September 2013

**Figure 3: The comparison of cross-sections from the simulation results to the field measurements after 2013 in Scenario PP.**

Figure 6 b and c need labels on axis.
We added labels in both subgraphs.

[Figure]

**Figure 4: (a) Simulated geomorphic changes over time for three scenarios; (b) the affected area of deposition and erosion for three scenarios; (c) columnar distribution of different erosion and deposition levels.**

---

## Author Response (AR3)

**Point-by-point replies**

Title: " An assessment of Short-medium Term Intervention Using CAESAR-Lisflood in Post-earthquake Mountainous Area"

Authors: Di Wang, Ming Wang, Kai Liu, Jun Xie

Manuscript Number: nhess-2022-195

We would like to thank the editor for the thorough reading of the manuscript and giving an opportunity to improve the manuscript. The texts with blue font are the editor's original comments, the texts with black font are authors' responses.

Thanks for revising the manuscript according to the reviewer's minor suggestions. I found the paper's scientific content to be robust and relevant to NHESS readers. But in reviewing the manuscript, I noticed many sentences that need polishing for their English. The readers may find the text understandable, but it requires additional effort and the language is not fluent. Please revise the manuscript for its language, preferably with a native English speaker familiar with the relevant jargon and terminology, and I will evaluate it again afterward. I am looking forward to receiving the revised manuscript.

We have revised the manuscript carefully for proper English language, grammar, punctuation, spelling, and overall style according to the highly qualified native English speaking editor at AJE. The editing certificate is shown below. All of the changes in the manuscript are formatted in purple and underline shown in the track changes file.

[Figure]

**AJE** | **Editing Certificate**

This document certifies that the manuscript

**Assessment of Short-medium Term Intervention Effects Using CAESAR-Lisflood in Post-earthquake Mountainous Area**

prepared by the authors

**Di Wang, Ming Wang, Kai Liu, Jun Xie**

was edited for proper English language, grammar, punctuation, spelling, and overall style by one or more of the highly qualified native English speaking editors at AJE.

This certificate was issued on **February 6, 2023** and may be verified on the AJE website using the verification code **202A-6E97-FBE6-9EFF-89E6**.

Neither the research content nor the authors' intentions were altered in any way during the editing process. Documents receiving this certification should be English-ready for publication; however, the author has the ability to accept or reject our suggestions and changes. To verify the final AJE edited version, please visit our verification page at aje.com/certificate. If you have any questions or concerns about this edited document, please contact AJE at support@aje.com.

AJE provides a range of editing, translation, and manuscript services for researchers and publishers around the world. For more information about our company, services, and partner discounts, please visit aje.com.

Additionally, we adjusted the legends and more details in Fig. 8 and Fig.11. The newest versions are shown as follows.

[Figure]

**Figure 1: The depth of deposited sediment in the dams' placements.**

[Figure]

**Figure 2: Rainfall input of ten years and relative efficiency of sediment intervention measures. (a) Relative efficiency changes over ten years (the grey region highlighting stage III, and the dashed lines indicate the linear fitting curves); (b) Rainfall downscaled from NEX-GDDP (NASA Earth Exchange Global Daily Downscaled Projections) product.**

Thank you again for the opportunity to be considered for publication in NHESS.

---

## Author Response (AR4)

**Point-by-point replies**

Title: " An assessment of Short-medium Term Intervention Using CAESAR-Lisflood in Post-earthquake Mountainous Area"

Authors: Di Wang, Ming Wang, Kai Liu, Jun Xie

Manuscript Number: nhess-2022-195

We would like to thank the editor for the thorough reading of the manuscript and giving another opportunity to improve the manuscript. The texts with blue font are the editor's original comments, the texts with black font are authors' responses.

I appreciate you proofreading the manuscript and revising the figures. After the revisions, the manuscript reads better, but **further English proofreading** is required. There are still a few **grammatical errors** in the manuscript, in addition to the fact that many of the **sentences are complex** to understand and the **language does not flow well**. The problem can be solved by having a native English speaker proofread the document. Moreover, the text **does not use the correct jargon** used by geomorphologists and geohazard specialists. Consequently, it is essential that the proofreading be performed by a specialist in the field who can **adjust the terminology** according to the scientific community standards. The manuscript cannot be accepted for publication in NHESS in its current state. I have placed the manuscript again in a status of "minor revisions" and am giving you another opportunity to revise the document. I am looking forward to receiving the revised manuscript..

We have revised the manuscript carefully to make sure we used the right grammar. We alse refined sentences and modified some of the teminology used by geomorphologists. For the complex sentences, we removed unnecessary phrases and split up the sentence structure to make it clearer and more fluent. We have deleted some texts irrelevant to the important results. In addition, we updated the jargons usede by the specilists and revised the manuscript accordingly. Some typical terms we have modified are listed bellow.

| No. | Original terminology | Revised terminology |
|-----|----------------------|---------------------|
| 1 | checking dams | check dams |
| 2 | vegetation revetments | vegetated slopes/slope protection with vegetation |
| 3 | engineering | geotechnical engineering |
| 4 | biological measures | ecological engineering |
| 5 | indicators | indices |
| 6 | cooperative control measures | comprehensive control measures |
| 7 | landscape changes | landform changes |
| 8 | deposit area | deposition area |

Moreover, we revised the legends in Fig.1, Fig.6 and Fig.7 according to the updated terminology. Thank you again for the opportunity to be considered for publication in NHESS. The latest editing certificate is shown below. All of the changes in the manuscript are formatted in red and underline shown in the track changes file.

[Figure]

**Editing Certificate**

This document certifies that the manuscript

**Assessment of Short-medium Term Intervention Effects Using CAESAR-Lisflood in Post-earthquake Mountainous Area**

prepared by the authors

**Di Wang, Ming Wang, Kai Liu, Jun Xie**

was edited for proper English language, grammar, punctuation, spelling, and overall style by one or more of the highly qualified native English speaking editors at AJE.

This certificate was issued on **February 28, 2023** and may be verified on the AJE website using the verification code **202A-6E97-FBE6-9EFF-89E6**.

[Figure]

Neither the research content nor the authors' intentions were altered in any way during the editing process. Documents receiving this certification should be English-ready for publication; however, the author has the ability to accept or reject our suggestions and changes. To verify the final AJE edited version, please visit our verification page at aje.com/certificate. If you have any questions or concerns about this edited document, please contact AJE at support@aje.com.

AJE provides a range of editing, translation, and manuscript services for researchers and publishers around the world. For more information about our company, services, and partner discounts, please visit aje.com.